# Focal adhesions are controlled by microtubules through local contractility regulation

Julien Aureille [1✉], Srinivas S Prabhu[1], Sam F Barnett[1], Aaron J Farrugia[1], Isabelle Arnal[2], Laurence Lafanechère [3], Boon Chuan Low [1], Pakorn Kanchanawong[1,4], Alex Mogilner[5] & Alexander D Bershadsky [1,6✉]

## Abstract

**Microtubules regulate cell polarity and migration via local activation of focal adhesion turnover, but the mechanism of this process is insufficiently understood. Molecular complexes containing KANK family proteins connect microtubules with talin, the major component of focal adhesions. Here, local optogenetic activation of KANK1-mediated microtubule/talin linkage promoted microtubule targeting to an individual focal adhesion and subsequent withdrawal, resulting in focal adhesion centripetal sliding and rapid disassembly. This sliding is preceded by a local increase of traction force due to accumulation of myosin-II and actin in the proximity of the focal adhesion. Knockdown of the Rho activator GEF-H1 prevented development of traction force and abolished sliding and disassembly of focal adhesions upon KANK1 activation. Other players participating in microtubule-driven, KANK-dependent focal adhesion disassembly include kinases ROCK, PAK, and FAK, as well as microtubules/focal adhesion-associated proteins kinesin-1, APC, and αTAT. Based on these data, we develop a mathematical model for a microtubule-driven focal adhesion disruption involving local GEF-H1/RhoA/ROCK-dependent activation of contractility, which is consistent with experimental data.**

**Keywords** KANK1; GEF-H1; Myosin-II; iLID Optogenetics; Focal Adhesion Mechanosensitivity
**Subject Category** Cell Adhesion, Polarity & Cytoskeleton

## Introduction

Microtubules are ubiquitous cytoskeletal elements. In many cell types, they are responsible for polarity of cell shape and cell directional migration (Etienne-Manneville, 2013; Meiring et al, 2020; Small et al, 2002). This function of microtubules depends on their interaction with integrin-mediated cell-matrix adhesions. In this study, we focus on microtubule interaction with a special type of integrin adhesions known as focal adhesions.

Focal adhesions are associated with the contractile actomyosin cytoskeleton and transduce forces generated by associated actomyosin structures (stress fibers) to the extracellular matrix. A variety of experimental approaches based on the use of elastic deformable substrate (traction force microscopy) clearly demonstrated that many cell types exert forces applied to the substrate at discrete sites corresponding to focal adhesions (Balaban et al, 2001; Beningo et al, 2001; Schmitt et al, 2024). At the same time, the focal adhesions are also sensing the forces: they disassemble when myosin-IIA driven actomyosin contractility is inhibited and grow in size when the contractility increases or external force is applied (Balaban et al, 2001; Chen et al, 2013; Even-Ram et al, 2007; Lavelin et al, 2013; Riveline et al, 2001; Vicente-Manzanares et al, 2007; Zheng et al, 2021).

Disruption of microtubules results in increase of focal adhesion size in a myosin-II-dependent fashion (Bershadsky et al, 1996). The mechanism underlying this phenomenon is based on activation of Rho and Rho kinase (ROCK) upon microtubule disruption (Liu et al, 1998; Ren et al, 1999). This Rho activation occurs due to release of a microtubule-associated Rho GEF, GEF-H1 and its activation (Chang et al, 2008; Krendel et al, 2002). Indeed, GEF-H1 knockdown abolished the growth of focal adhesions upon microtubule disruption (Chang et al, 2008; Rafiq et al, 2019).

While the growth and shrinking dynamics of microtubules is faster than turnover of focal adhesions, microtubules remain dynamically linked to focal adhesions through a complex protein network. The main players in microtubule coupling to focal adhesions are the KANK family proteins, which bind to a major focal adhesion component, talin, via their N-terminal KN domains (Bouchet et al, 2016; Sun et al, 2016). At the same time, the central coiled-coil domains of KANK proteins bind a liprin-β1, a component of membrane-associated complex that include also liprin-α, ELKS, and LL5β (Meiring et al, 2020). LL5β in turn binds the microtubule end tracking proteins CLASP1/2 (Lansbergen et al, 2006). In addition, ankyrin repeat domain at the C-terminus of KANK binds KIF21A kinesin, which is also located at the microtubule plus end (Bouchet et al, 2016; van der Vaart et al,

[1]Mechanobiology Institute, National University of Singapore, Singapore, Singapore. [2]Grenoble institute of Neuroscience, University Grenoble Alpes, INSERM U1216, Grenoble, France. [3]University Grenoble Alpes, INSERM U1209, CNRS UMR5309, Institute for Advanced Biosciences, Grenoble, France. [4]Department of Biomedical Engineering, National University of Singapore, Singapore, Singapore. [5]Courant Institute and Department of Biology, New York University, New York, USA. [6]Department of Molecular Cell Biology, Weizmann Institute of Science, Rehovot, Israel. ✉E-mail: Julien.aureille@univ-grenoble-alpes.fr; mbiba@nus.edu.sg

2013). While the existence of ternary complexes between talin, KANK, and liprin-β1 or between talin, KANK, and KIF21A was not directly demonstrated, it was shown that the double knockdown of KANK1 and KANK2 abolished the microtubule targeting to focal adhesions (Rafiq et al, 2019).

We have shown previously that disruption of the coupling between microtubules and focal adhesions by either knockdown of KANK proteins or by overexpression of KN domain, which interferes with the binding of endogenous KANK to talin, reproduces the effect of total disruption of microtubules (Rafiq et al, 2019). The GEF-H1 released from microtubules apparently undergoes activation and, in turn, activates Rho/ROCK cascade resulting in formation of numerous myosin-II filaments. This in turn, led to increase of focal adhesion sizes similarly to that observed upon total microtubule disruption (Rafiq et al, 2019). These data are consistent with previously published results showing that depletion of other elements in the microtubule-focal adhesion link such as CLASPs (Stehbens et al, 2014) or EB1 (Yang et al, 2017) as well as ELKS (ERC1) (Astro et al, 2016), results in suppression of focal adhesion turnover. Thus, both microtubule disruption and their disconnection from focal adhesions induce focal adhesion growth in GEF-H1 and myosin-II-dependent manner.

In turn, the microtubule outgrowth after washing out a microtubule disrupting drug leads to transient disassembly of the focal adhesions (Ezratty et al, 2005). Using this experimental model, several candidate proteins related to the microtubule-mediated disassembly of focal adhesions were proposed (Bhatt et al, 2002; Ezratty et al, 2005; Juanes et al, 2019; Kenific et al, 2016; Krylyshkina et al, 2002; Yue et al, 2014). However, this approach only allowed assessment of the alteration of focal adhesions in the entire cell rather than the effect of microtubules targeting to individual focal adhesions. In addition, since the process of microtubule outgrowth is not entirely synchronous, the time-course of the microtubule-driven focal adhesion disassembly in this experimental system is difficult to follow. Finally, these studies did not assess the changes in myosin-II contractility that, as we mention above, are critically important for the understanding of microtubule interactions with focal adhesions.

In the present study, we developed an optogenetic approach which permitted us to target microtubules to selected individual focal adhesions. This approach is based on using the iLID optogenetic system (Guntas et al, 2015) which, upon blue light illumination, restored the link between two halves of KANK molecule promoting the KANK-mediated contacts between microtubules and illuminated adhesions. We have shown that microtubule targeting indeed results in focal adhesion disassembly and analyzed this process in detail.

We have found that the process of microtubule-driven focal adhesion disassembly requires GEF-H1-dependent local activation of myosin-II filament formation. This burst of actomyosin contractility triggers the disruption of the adhesion by inducing its sliding during which the focal adhesion undergoes rapid disassembly. We have identified the microtubule- and focal adhesion-associated proteins involved in this process and showed that some of them mediate the burst of contractility triggering the focal adhesion sliding while others likely weakened the adhesion strength making the sliding possible. We propose a plausible quantitative model of the entire process of interactions of microtubules with focal adhesion leading to focal adhesion disassembly.

# Results

## Disassembly of focal adhesions upon induction of KANK-mediated link

We created an iLID system-based optogenetic construct (Guntas et al, 2015) of KANK1 protein (OptoKANK) which consists of (i) KANK1 talin binding domain (KN) fused with mApple at its N-terminus and LOV2ssrA at the C-terminus and (ii) remaining part of KANK1 molecule (ΔKN) fused with SSpB at the N-terminus and mEmerald at the C-terminus (Fig. 1A). The constructs were designed in a similar way as the constructs for optogenetic activation of talin described previously (Yu et al, 2020).

Western blotting using KANK1 antibody showing the levels of SSpB-ΔKN-mEmerald expression together with that of endogenous KANK1 in the same samples allowed us to roughly estimate that OptoKANK construct expression in typical experiments was comparable with that of endogenous KANK1 (Appendix Fig. S1A). This level of overexpression did not induce apoptotic cell death as demonstrated by caspase 3 activation assay (Appendix Fig. S1B). At the same time, OptoKANK expression at this level seems to be sufficient to displace endogenous KANK1 and thereby disconnect microtubules from the focal adhesions. This can be inferred from the fact that the cells expressing OptoKANK demonstrated the higher level of myosin II-driven contractility, manifested by increased size of focal adhesions, than control cells (Appendix Fig. S1C). This increase of focal adhesion size was similar to that in the cells with KANK1 knockdown or overexpression of KANK-KN domain (Rafiq et al, 2019). We also observed the increase of phosphorylated FAK (pY397) level in both KANK1/2 knockdown cells and in these cells expressing OptoKANK construct (Appendix Fig. S1D).

Total Internal reflection fluorescence (TIRF) microscopy of distribution of mApple-KN-LOV2ssrA, and SSpB-ΔKN-mEmerald in HT1080 cells together with focal adhesion marker vinculin-mIFP revealed that KN-LOV2ssrA localized to focal adhesions, while SSpB-ΔKN did not, and demonstrated weak surface fluorescence (Fig. 1B). Illumination of a small circular area containing a focal adhesion did not affect localization of KN but led to the relocation of ΔKN into illuminated region overlapping with the focal adhesion (Fig. 1B).

Assessment of the vinculin fluorescence intensity revealed that blue illumination of mature peripheral focal adhesion in cells containing complete OptoKANK (KN + ΔKN) induced centripetal sliding of the focal adhesion accompanied by a decrease of vinculin fluorescence (Fig. 1C; Movie EV1). The total disappearance of the focal adhesion was observed within about 10 min. The focal adhesions in cells expressing incomplete OptoKANK (either only KN or ΔKN) were not affected by the illumination (Fig. 1C).

We selected for our experiments well-developed mature focal adhesions. Besides technical convenience, justification for this is that usually focal adhesions undergo maturation earlier than microtubule ends approach them. Indeed, the zone of the highest frequency collisions between microtubule tips and focal adhesion was shown to be located between 20 to 40 μm behind the leading

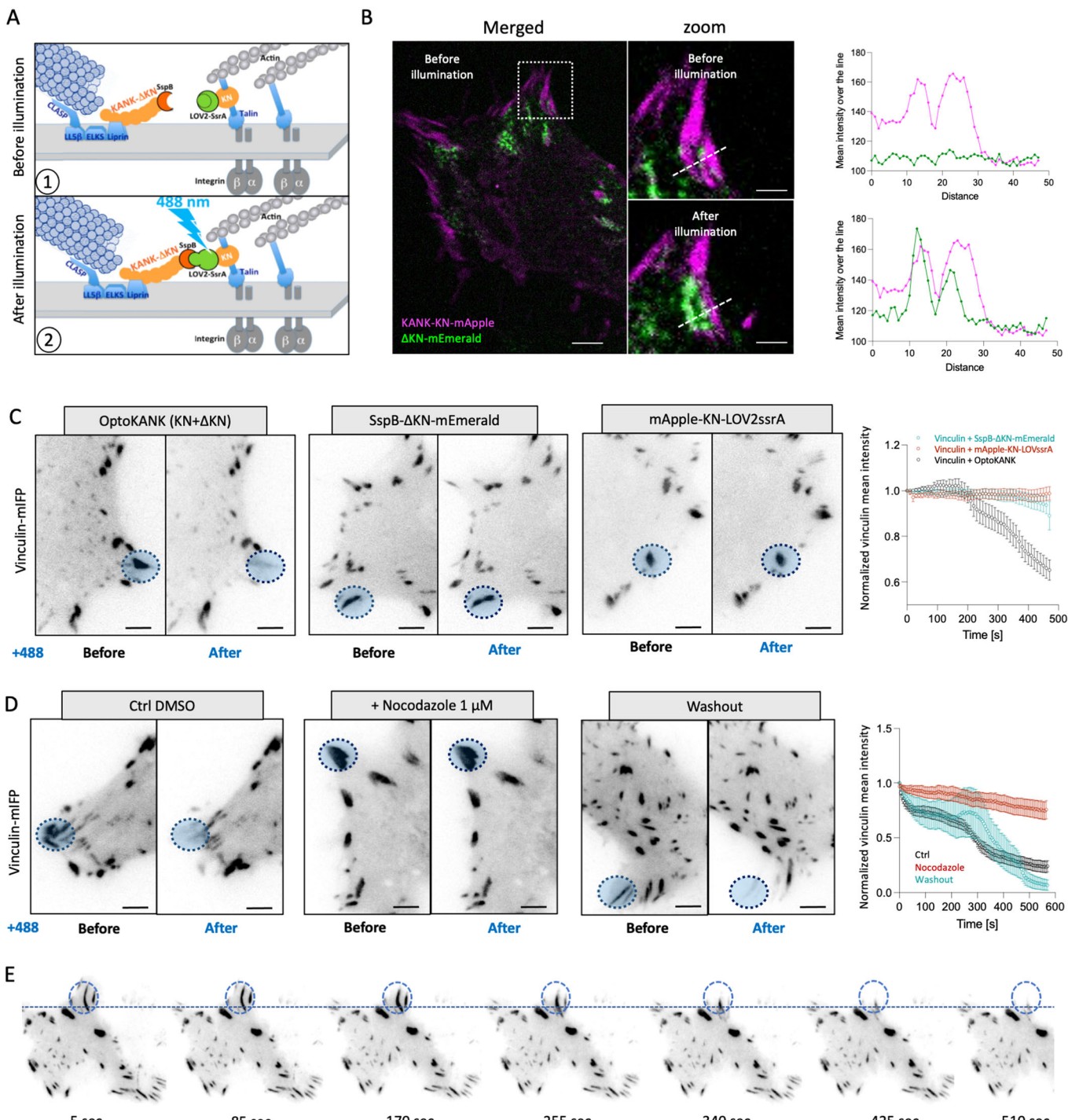

edge, where the adhesions approach maximal size (Rid et al, 2005). Thus, our optogenetic targeting of microtubules to mature adhesions reflects the typical physiological situation.

While majority of HT1080 cells formed only peripheral adhesions, in some cells the adhesion can be seen also in the central region. These centrally-located focal adhesions were less sensitive to OptoKANK activation than peripheral ones. They sometimes demonstrated sliding but not complete disassembly during the period of observation (Appendix Fig. S1E,F). All further

experiments described in this study are related to peripheral focal adhesions.

The disassembly of focal adhesions upon activation of OptoKANK can be observed also in the cells depleted of endogenous KANK1 and KANK2. The western blot analysis with antibodies to KANK1 and KANK2 as well as antibody to mApple visualizing mApple-KN-Lov2ssrA, showed that both KANK1 and 2 were almost entirely depleted (Appendix Fig. S1G). Nevertheless, the blue light illumination of focal adhesions visualized by vinculin-

Figure 1.   Disassembly of focal adhesions upon induction of KANK-mediated link.

(A) Schematic illustration of OptoKANK activation. (1) In dark state the C-terminal helix of the LOV2 domain prevents the SsrA peptide to bind its binding partner SspB. (2) Upon blue illumination (488 nm) LOV2 changes its conformation allowing SsrA to bind SspB leading eventually to the functional activation of OptoKANK. (B) Representative images of OptoKANK dimerization in HT1080 cells upon photoactivation. Before illumination, KANK-ΔKN-mEmerald (ΔKN) is diffused within the cell. After 150 s of blue light illumination, ΔKN is recruited to KN. The graphs show the KN and ΔKN mean intensities over the line (dotted white line) in the representative images, before and after photoactivation of KANK (scale bar 6 μm and 3 μm). (C) Representative images of Vinculin-mIFP-transfected HT1080 cells carrying the full OptoKANK constructs (KN + ΔKN), or either the KN construct or the ΔKN construct, before and after 8 min of blue light illumination on the focal adhesion (blue dotted circle). Graph shows the average of mean vinculin normalized as a fraction of initial intensity of illuminated focal adhesion over the time in these three conditions. (Data are presented as mean ± s.e.m.; n = 19 cells minimum; scale bar 10 μm). See also movie EV1. (D) Representative images of Vinculin-mIFP-transfected HT1080 cells carrying the full OptoKANK constructs (KN + ΔKN) before and after 540 s of blue light illumination on the focal adhesion (blue dotted cercle) for cells in control (Ctrl DMSO), treated with 1 μM of nocodazole, and after the nocodazole washout. Graph shows the vinculin intensities normalized as a fraction of initial intensity of illuminated focal adhesions over the time in these three conditions. (Data are presented as mean ± s.e.m.; n = 8 cells from two independent experiments). See also movie EV2. (E) Time-course of the focal adhesion disassembly after nodazole washout upon OptoKANK activation (blue dotted circle). See also movie EV3.

mIFP resulted in their sliding and disassembly in exactly similar manner as in wild-type cells containing endogenous KANK1 and KANK2 (Appendix Fig. S1H). Thus, the effect of activation of OptoKANK is not the result of its interaction with endogenous KANK1/2.

To check whether this effect was due to targeting of microtubules to focal adhesions we compared the effect of focal adhesion illumination in non-treated cells expressing OptoKANK with that in cells treated with nocodazole for 2 h and cells treated with nocodazole and then washed out for 2 h. We found that disruption of microtubules with nocodazole abolished the effect of illumination on the focal adhesion integrity while washing the drug out restored the disruptive effect of illumination (Fig. 1D,E; Movies EV2 and EV3). Thus, the optogenetics driven formation of KANK1-mediated link between microtubules and focal adhesions resulted in sliding and disassembly of focal adhesion in a microtubule-dependent fashion.

## Targeting of microtubules to the focal adhesion area upon local OptoKANK activation

We further assessed how the number of microtubule tips in the focal adhesion area changes upon local photoactivation of OptoKANK. In the first series of experiments, the microtubule tips were labeled with EB3-mIFP co-transfected with the Opto-KANK constructs. The focal adhesions were identified by labeled KN domain (mApple-KN-LOV2ssrA) (Fig. 1B), which as mentioned above, always co-localized with vinculin-labeled focal adhesion. Manual counting of EB3 "comets" overlapping with focal adhesion area revealed approximately a 25% increase in the number of microtubule tips already at 30 s following the onset of illumination (Fig. 2A). There were no changes in the average number of microtubule tips at non-illuminated focal adhesions in the same cell during this time interval (Fig. 2A).

To obtain further quantitative data, we plated cells on a micropatterned substrate with circular fibronectin-coated islands of 4.5 μm in diameter. On this substrate, cells could only form focal adhesions within these adhesive islands (usually one to three focal adhesion per island—Appendix Fig. S2A). We further measured the dynamics of fluorescence intensity of EB3-mIFP over time in both illuminated and non-illuminated islands. We illuminated three islands per cell leaving others non-illuminated. The analysis of the time course of EB3 fluorescence revealed an apparent fluctuation in both illuminated and non-illuminated areas (Fig. 2B,C; Appendix

Fig. S2B; Movie EV4). Nevertheless, in all cases the fluorescence intensity of EB3 per island was higher in the illuminated islands with photoactivated OptoKANK (Fig. 2B,C; Appendix Fig. S2B) consistent with measurements shown in Fig. 2A.

The EB1 and EB3 proteins recognize the growing plus ends of microtubules that contain GTP caps but not the shrinking ends (Maurer et al, 2012). Therefore, to fully quantify the microtubules targeting to focal adhesions, we used SiR-Tubulin labeling (Lukinavicius et al, 2014). These experiments revealed a stereotypic dynamic of the intensity of SiR-Tubulin fluorescence overlapping with focal adhesions upon OptoKANK activation by illumination. The SiR-Tubulin fluorescence increased in the first 60 s following onset of illumination and then decreased below the level typical for non-illuminated focal adhesions (Fig. 2D,E; Movie EV5). These results suggest that activation of OptoKANK promoted targeting of microtubules to focal adhesions followed by microtubule withdrawal.

The microtubules recruited to the focal adhesions by Opto-KANK illumination preserved their transport functions. As it has been demonstrated in some cellular systems, microtubules deliver exocytotic vesicles to the plasma membrane-associated secretory sites in a KANK-dependent fashion (Noordstra and Akhmanova, 2017; Noordstra et al, 2022). Normally, accumulation of the new membrane at these sites is balanced by the local endocytosis. We expected that in the presence of endocytosis inhibitor, dynasore (Macia et al, 2006), the accumulation of the functional microtubule tips by OptoKANK illumination will augment the local delivery of the new membrane. Indeed, we have found that illumination of focal adhesions in dynasore-treated cells containing OptoKANK construct resulted in formation of membrane blebs at the illuminated sites (Appendix Fig. S2C; Movie EV6).

## Dynamics of myosin-II filaments and traction forces upon local OptoKANK activation

Observations of myosin-II filaments labeled with myosin light chain fused with near-infrared fluorescent protein (MLC-mIFP) revealed that activation of OptoKANK by local illumination of focal adhesions resulted in rapid accumulation of myosin filaments in the proximity of focal adhesions (Fig. 3A; Movie EV7). Myosin-II filaments fluorescence signal appeared in a centripetal direction from the focal adhesions with a maximum at approximately 4 μm distance from the focal adhesion proximal end (Fig. 3B). Time course analysis of the burst of myosin-II filaments accumulation

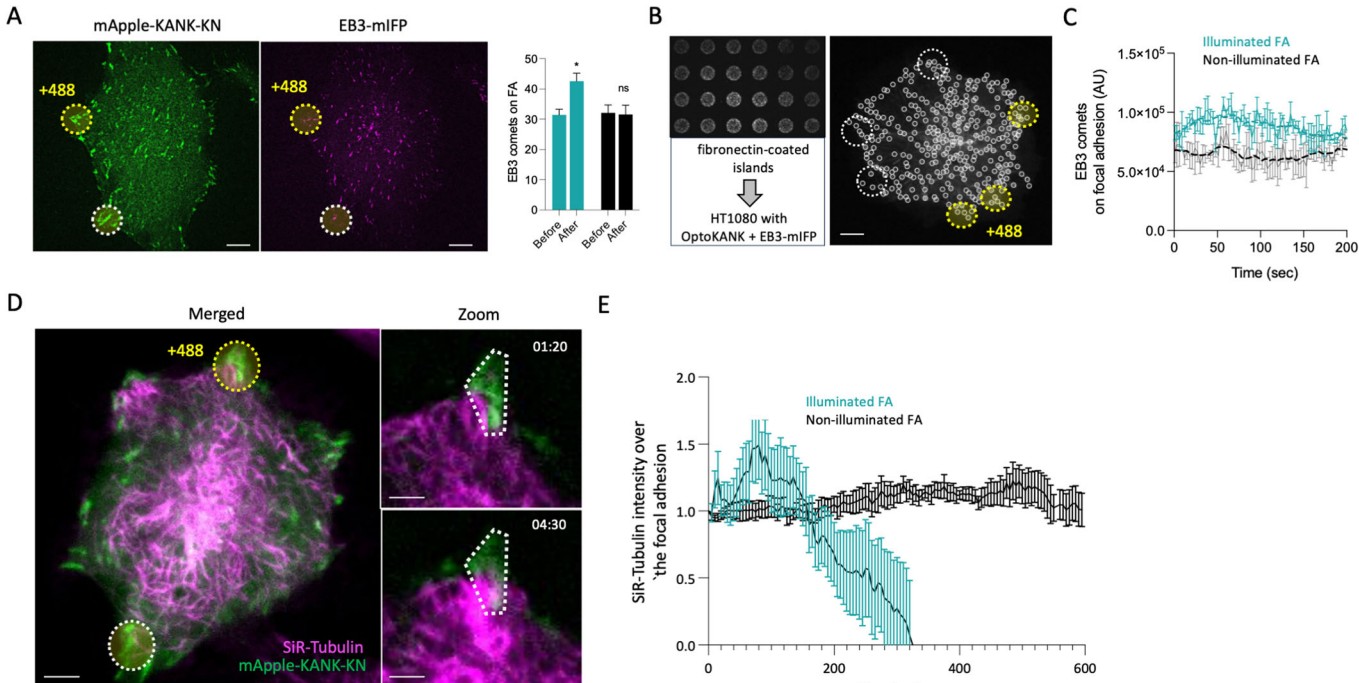

**Figure 2. Augmented targeting of microtubules to the focal adhesion area upon local OptoKANK activation.**

(A) Representative images HT1080 cells transfected with OptoKANK and EB3-mIFP. Focal adhesions are visualized with KN construct. The yellow dotted line represents the blue light illumination area. The white dotted line represents the control (no illumination). Graphs show the average number of EB3 comets reaching the illuminated and non-illuminated focal adhesions before and after 30 s of blue light illumination. (Data are presented as mean ± s.e.m.; n = 9, t-test; *p-value < 0.05; scale bar 10 μm). (B) Representative image of fibronectin-coated islands on which the HT1080 cells transfected with OptoKANK and EB3-mIFP were plated. EB3-mIFP marker was used to assess the number of EB3 comets that reach photoactivated (yellow dotted circles) and non-photoactivated (white dotted circles) focal adhesions which were visualized using the KN construct (scale bar 10 μm). On the right panel, representative image processed with the U-Track 2 script (see Methods) that was used to quantify the number of EB3 comets reaching the different regions of interest. (C) Graph shows the integrated EB3 comet fluorescence after processing with U-Track2 for photoactivated and non-photoactivated focal adhesions over the time. (Data are presented as the mean ± s.e.m. of integrated EB3 fluorescence. n = 3 focal adhesions per cell; scale bar 10 μm). (D) Representative image of OptoKANK-transfected HT1080 cell after 80 s and 270 s of blue light illumination of the focal adhesion (dotted yellow line). SiR-Tubulin at 250 nM added 3 h prior the experiments was used to visualize microtubule. The white dotted line encloses non-illuminated focal adhesion used as control. In the zoomed boxes, white dotted lines enclose the regions of interest where illuminated focal adhesions are labeled by the mApple-KANK-KN (green) and microtubules are labeled by SiR-Tubulin (magenta) (scale bar 10 μm on the left, and 5 μm for the zoom boxes). See also movie EV5. (E) Graph shows the mean SiR-Tubulin intensity at the photoactivated and non-photoactivated focal adhesions over the time (Data are presented as mean ± s.e.m.; n = 18 biological replicates from 2 independent experiments).

revealed that the myosin filament density approaches the maximum in about 100 s following the onset of illumination (Fig. 3C,D). The accumulation of F-actin visualized by SiR-Actin (Lukinavicius et al, 2014) was observed at the same time and location (Appendix Fig. S3; Movie EV8). Comparison of the dynamics of myosin filaments accumulation with those of microtubule and vinculin density in focal adhesion area showed that the appearance of myosin filaments at the proximal end of focal adhesion coincided with withdrawal of microtubules that were attracted to focal adhesions shortly after the onset of illumination (Fig. 3D). In turn, the moment when the density of myosin filaments approached the maximum preceded the process of focal adhesion disassembly (Fig. 3D).

To check if accumulation of myosin filaments affects the traction forces applied through the focal adhesions, we plated cells on the array of elastic micropillars (Gupta et al, 2015) and assessed the time-course of micropillar deflections in both illuminated and non-illuminated areas (Fig. 3E). Measurements of traction force at the focal adhesion area (approximately 4 pillars per focal adhesion) over the time revealed a significant increase of traction force upon

illumination-induced OptoKANK activation compared to the non-illuminated area (Fig. 3E).

Accumulation of myosin filaments in proximity to focal adhesions and development of traction forces appeared to be critically important for the disassembly of focal adhesions upon forced targeting of microtubules by OptoKANK activation. Treatment with Y27632 abolished the traction force increase and prevented the sliding and disassembly of focal adhesions (Fig. 3F). Of note, the treatment with low dose of Y27632 in these experiments lasting only for the period of illumination, was used only to prevent the increase of traction forces developed due to myosin II filaments accumulation upon microtubule targeting to focal adhesions. More pronounced treatment with this drug which resulted in complete annulation of traction forces led to disassembly of focal adhesions as was well established in numerous previous studies (Lavelin et al, 2013; Riveline et al, 2001).

Thus, targeting of microtubules to focal adhesions by Opto-KANK activation results in transient accumulation of myosin-II filaments near the proximal end of the focal adhesions and bursts of traction force, which are required for focal adhesion disassembly.

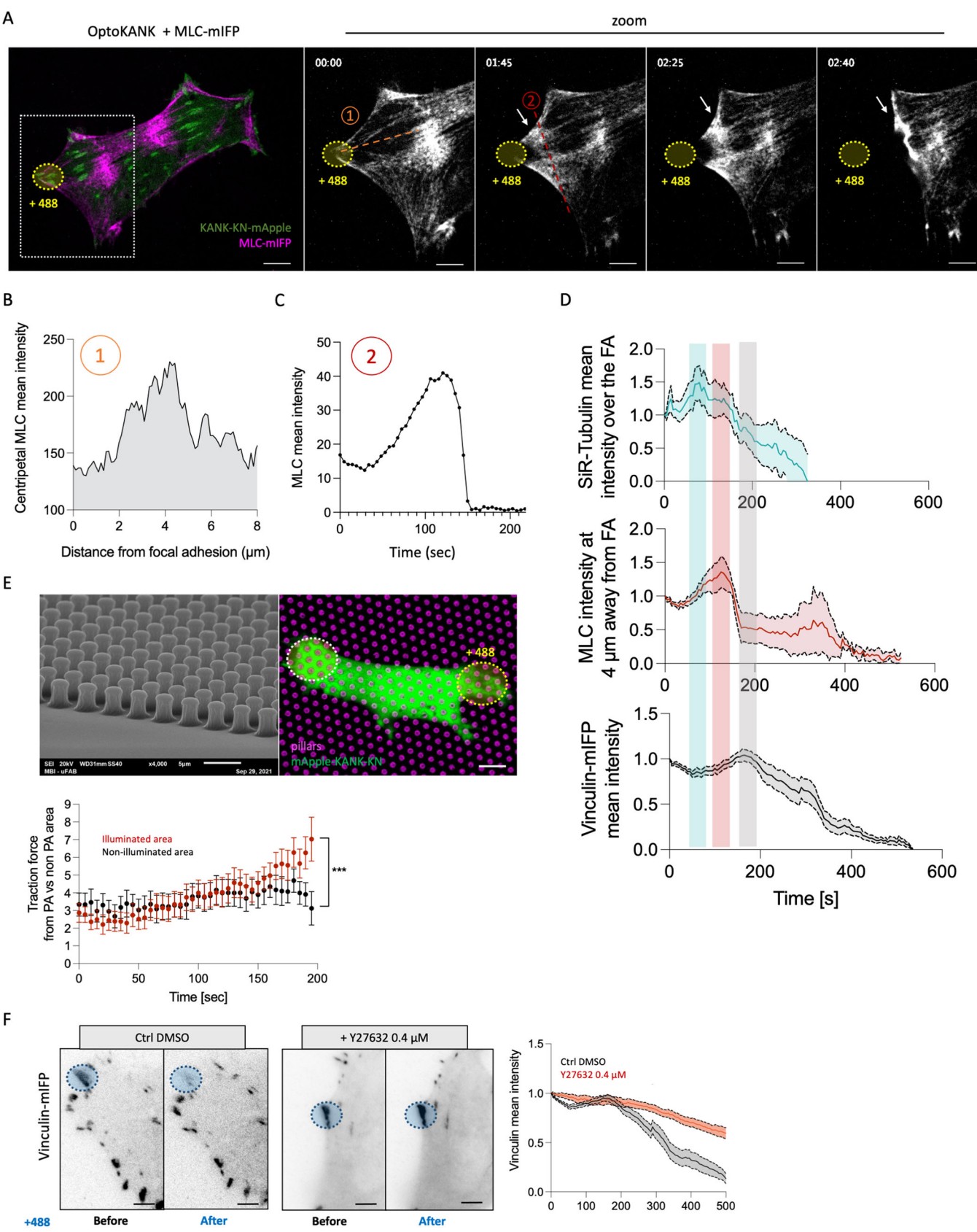

◄ **Figure 3.  Dynamics of myosin-II filaments and traction forces upon local OptoKANK activation.**

(A) Representative image of OptoKANK-transfected HT1080 cell at different time points of blue light illumination of the focal adhesion (yellow dotted line). MLC-mIFP was used to assess the myosin-II dynamics upon OptoKANK activation. The line scan (1) was used to measure the MLC-mIFP intensity in the vicinity (4 μm away in centripetal direction) of the proximal end of the photoactivated focal adhesion shown in (B). The curve (2) shows the MLC-mIFP mean intensity in the proximity of the photoactivated focal adhesion over the time (C) (scale bar 10 μm). See also movie EV7. (D) Time courses of the microtubule intensity overlapping with focal adhesions (SiR-Tubulin—blue rectangle corresponds to maximum value), MLC intensity at 4 μm from the proximal end of the illuminated focal adhesion (MLC-mIFP mean intensity— red rectangle corresponds to maximum value) and vinculin mean intensity in focal adhesion (Vinculin-mIFP mean intensity—gray rectangle corresponds to maximum value). Note that increase of the microtubule intensity upon illumination succeeded by its drop, which coincide with the increase of MLC intensity in proximity of the focal adhesion, which preceded the drop in vinculin intensity. (Data are presented as mean ± s.e.m.; Vinculin-mIFP, $n = 8$; MLC intensity, $n = 6$; SiR-Tubulin, $n = 18$. Each dataset is obtained in at least two independent experiments). (E) Representative images of OptoKANK-transfected HT1080 cell on pillars. The illuminated and non-illuminated focal adhesions are encircled by yellow and blue dotted circles, respectively. The cells were plated on micropillars (magenta) for traction force assessment over the time. Graph shows the average traction force at the focal adhesion surrounding (based on measurement of deflection of about 3-4 pillars per focal adhesion) upon illumination-induced OptoKANK activation compared to the traction forces in non-illuminated area. (Data are presented as mean ± s.e.m.; $n = 19$ pillars from 6 different cells; two-way ANOVA, ***$p < 0.001$; scale bar 5 μm). (F) Representative images of Vinculin-mIFP-transfected HT1080 cells carrying the OptoKANK constructs (KN + ΔKN) before and after 500 s of blue light illumination of the focal adhesion (blue dotted line) for control cells (Ctrl DMSO), and cells treated with 0.4 μM of Y27632. Graph shows the normalized mean vinculin intensity of illuminated focal adhesions over the time under these two conditions. (Data are presented as mean ± s.e.m.; $n = 18$ cells minimum from two independent experiments; scale bar 5 μm).

## Rho activation by GEF-H1 is required for focal adhesion disassembly upon microtubule targeting

Disruption of microtubules or their disconnection from integrin adhesions induces formation of myosin filaments due to release and activation of GEF-H1 and subsequent activation of Rho and ROCK (Krendel et al, 2002; Rafiq et al, 2019). Thus, we decided to check whether transient activation of myosin filaments formation in the proximity of focal adhesions and subsequent focal adhesion sliding and disassembly upon OptoKANK activation also depends on GEF-H1 and Rho. We found that in GEF-H1 depleted cells activation of OptoKANK by the illumination of focal adhesions did not result in increase of traction force applied to the focal adhesions (Fig. 4A). Consistently the OptoKANK activation in GEF-H1 depleted cells did not trigger the sliding and disassembly of the focal adhesions (Fig. 4B; Movie EV9).

Further, we rescued the effect of GEF-H1 knockdown by activation of Rho. Treatment of OptoKANK expressing cells with Rho activator CNO3 increased the focal adhesion sizes but did not prevent the sliding and disassembly of focal adhesions upon OptoKANK activation by illumination (Fig. 4C). The treatment of OptoKANK expressing GEF-H1 knockdown cells with CNO3 during the focal adhesion illumination rescued the effect of GEF-H1 knockdown and resulted in sliding and disassembly of focal adhesions (Fig. 4C; Movie EV10). Thus, the burst of traction force and consequent disassembly of focal adhesions upon microtubule targeting depend on Rho activation by GEF-H1.

## OptoKANK-mediated disassembly of focal adhesions requires activity of FAK, PAK, Kinesin-1, αTAT1, and APC

Targeting of microtubules to focal adhesions by local activation of KANK provides a convenient experimental system to elucidate the function of different proteins in the microtubule-driven focal adhesion disassembly. As shown above, GEF-H1 and ROCK are important players in this process. Using siRNA-mediated knockdowns or/and pharmacological inhibitors, we screened a few candidates and established the involvement of several microtubule- and focal adhesion-associated proteins in the focal adhesion disassembly process induced by activation of KANK1-mediated link. Candidates were selected based on previous publications suggesting the involvement of these proteins in the regulation of focal adhesion turnover (Bhatt et al, 2002; Chorev et al, 2014; Even-Ram et al, 2007; Ezratty et al, 2005;

Juanes et al, 2020; Juanes et al, 2019; Krylyshkina et al, 2002; Pan et al, 2020; Seetharaman et al, 2022; Stehbens et al, 2014; Yue et al, 2014). The graph summarizing the results of these experiments is shown in Fig. 5A and Appendix Fig. S4. The readout in these experiments was the alteration of vinculin fluorescence intensity of the focal adhesions in cells expressing OptoKANK constructs upon illumination of this adhesion for 10 min. Among the 18 proteins screened, 7 were needed for the disassembly of focal adhesions induced by OptoKANK activation. Besides GEF-H1 and ROCK (Figs. 3F and 4B), the 5 proteins identified are the focal adhesion kinase FAK, the PAK family kinase members sensitive to FRAX1036 inhibitor (Ong et al, 2015), kinesin-1 as suggested by experiments with kinesore inhibitor (Randall et al, 2017), tubulin acetylase αTAT1 and microtubule-associated actin regulator adenomatous polyposis coli (APC) protein (Fig. 5A). The results implicating FAK, APC, and Kinesin-1 in microtubule-driven focal adhesion disassembly are consistent with previous publications (Ezratty et al, 2005; Hamadi et al, 2005; Juanes et al, 2017; Krylyshkina et al, 2002).

The suppressive effect of GEF-H1 knockdown on focal adhesion disassembly by OptoKANK activation can be canceled by activation of Rho (Fig. 4C), which suggests that the function of GEF-H1 in this process is indeed RhoA activation. To elucidate which other proteins identified by our screen are involved in Rho/myosin IIA activation after microtubule targeting, we checked whether activation of Rho by CNO3 can cancel the inhibitory effect of knockdowns/pharmacological inhibition of these proteins. We found that besides GEF-H1, the effect of PAK kinase inhibition can be abolished by CNO3 treatment (Fig. 5B) suggesting that PAK function in microtubule-driven focal adhesion disassembly is also related to Rho/myosin II activation in agreement with previous publication on PAK function (Kiosses et al, 1999). Effects of knockdown or/and pharmacological inhibition of other candidates (FAK, Kinesin-1, APC) were not rescued by CNO3 treatment (Fig. 5C) suggesting that functions of these proteins are beyond the myosin contractility activation after microtubule detachment from the focal adhesion.

## Working hypothesis and a theoretical model of the process of OptoKANK activation driven focal adhesion disassembly

Our results suggest that local activation of actomyosin contractility in the proximity of focal adhesions is a necessary step in the

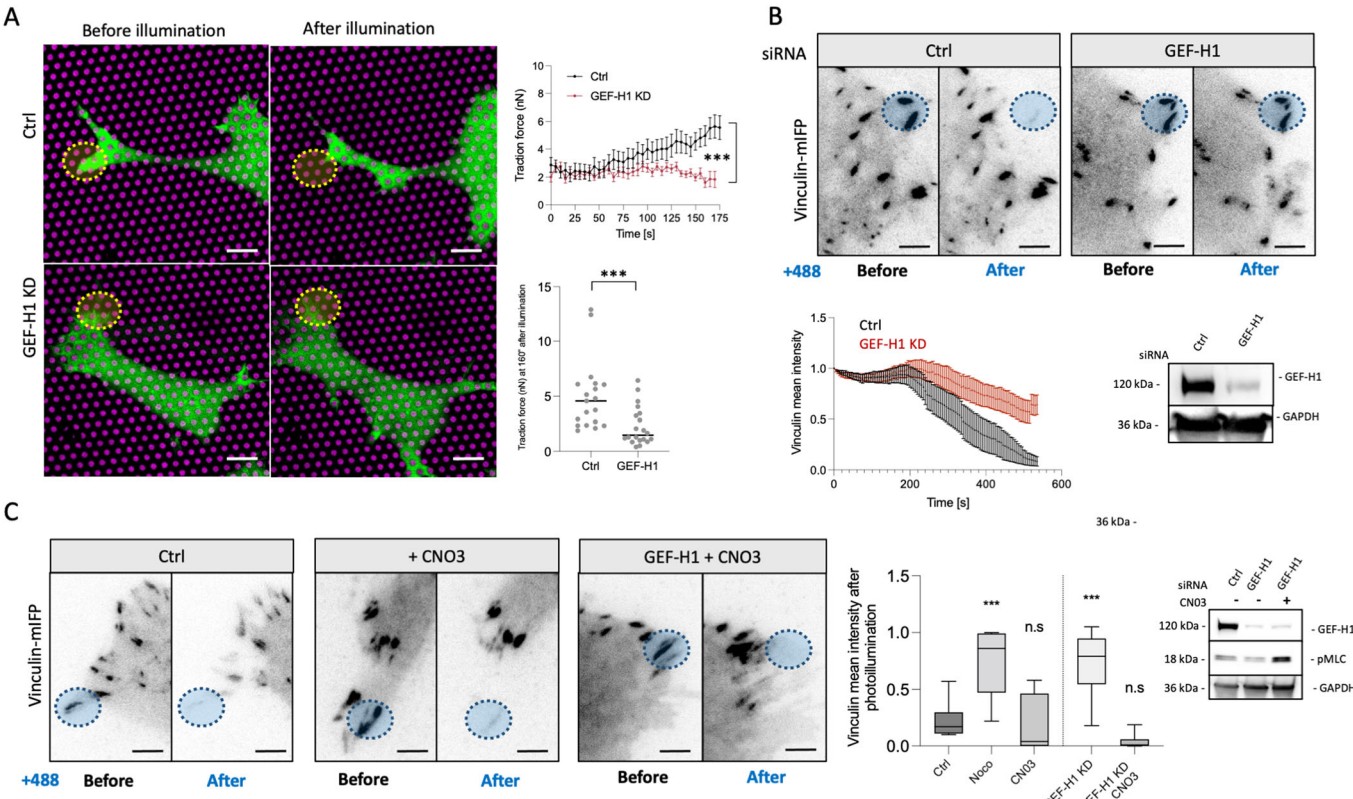

**Figure 4.  Rho activation by GEF-H1 is required for focal adhesion disassembly upon microtubule targeting.**

(**A**) Representative images of OptoKANK-transfected HT1080 cell before and after blue light illumination of focal adhesions encircled by yellow dotted line and plated on micropillars (magenta) for traction force assessment over the time under control (Ctrl) or GEF-H1 knockdown (GEF-H1 KD) conditions. Graph on the right shows the traction force of pillars in the region of interest over the time for control (Ctrl) or GEF-H1 depleted cells (GEF-H1 KD). Traction force after 160 s of blue light illumination is shown in the graph bellow (Data are presented as mean ± s.e.m.; Ctrl, n = 19 from 6 different cells; GEF-H1 KD, $n = 20$ from 7 different cells; two-way ANOVA, ***$p < 0.001$; scale bar 5 μm). (**B**) Representative images of Vinculin-mIFP-transfected HT1080 cells carrying the OptoKANK constructs before and after 8′ of blue light illumination on the encircled focal adhesion (yellow dotted line) for control (Ctrl), or depleted of GEF-H1 (GEF-H1 KD) cells. Graph shows the normalized mean vinculin intensity of illuminated focal adhesions over the time under these two conditions (Data are presented as mean ± s.e.m.; $n = 20$ minimum from three independent experiments; scale bar 5 μm). Immunoblots of GEF-H1 and GAPDH in control and GEF-H1 depleted cells are shown in the black box. See also movie EV9. (**C**) Representative images of Vinculin-mIFP-transfected HT1080 cells carrying the OptoKANK constructs before and after blue light illumination on the encircled focal adhesion (blue dotted line) of control cells (Ctrl), cells treated with CNO3 and GEF-H1 depleted cells treated with CNO3 (GEF-H1 KD + CNO3). Graph shows the normalized vinculin intensity after the illumination for control cells (Ctrl), cells treated with 1 μM nocodazole (negative controle), cells treated with CNO3, and GEF-H1 depleted cells treated with CNO3 (Data are presented as box and whisker plot; Ctrl, $n = 18$ cells; Noco; $n = 12$ cells; CNO3, $n = 8$ cells; GEF-H1 KD, $n = 24$ cells; GEF-H1 KD + CNO3, $n = 7$ cells. Data are from two independent experiments, one-way ANOVA, ***$p < 0.001$; the center line denotes the median value (50th percentile) while the box contains the 25th to 75th percentiles of dataset. The whiskers mark the minimum and maximum percentiles; scale bar 5 μm). Immunoblots of GEF-H1, phospho-MLC, and GAPDH under these three conditions are in the black box. See also movie EV10.

OptoKANK activation-driven disassembly of a focal adhesion. Indeed, we have shown that this process depends on GEF-H1 and ROCK and detected accumulation of actomyosin and increase of traction force in the proximity of a focal adhesion preceding its disassembly. Without specific microtubule targeting the augmentation of force applied to the focal adhesion results in their growth rather than disassembly (Riveline et al, 2001), therefore we assume that targeting of microtubules to the focal adhesion weakens the adhesion, enabling the focal adhesion to slide upon activation of myosin driven traction force. During this sliding, the force experienced by the focal adhesion drops (Shemesh et al, 2005), triggering its disassembly. Thus, based on our results, the process of focal adhesion disassembly after OptoKANK activation consists of the following steps (Fig. 6A–E). (i) Accumulation of microtubule tips at focal adhesions triggering weakening of the adhesions but

not yet their disassembly (Fig. 6B); (ii) the detachment and withdrawal of microtubule tips accompanied by release and activation of GEF-H1 which, with the assistance of PAK, results in accumulation of actin and myosin II filaments in the vicinity of the proximal end of the focal adhesion and activation of traction force (Fig. 6C); (iii) detachment and sliding of the focal adhesion as a result of burst of traction force, leading to its disassembly (Fig. 6D,E).

To analyze whether this working hypothesis can explain the process of microtubule-driven focal adhesion disassembly in accordance with experimental results, we propose a simple physical model of this process. In the model, we simulate on the computer dynamics of (a) number of microtubules in contact with a focal adhesion, (b) number of KANK molecules on the focal adhesion, (c) myosin II filament density proximal to the focal adhesion, (d)

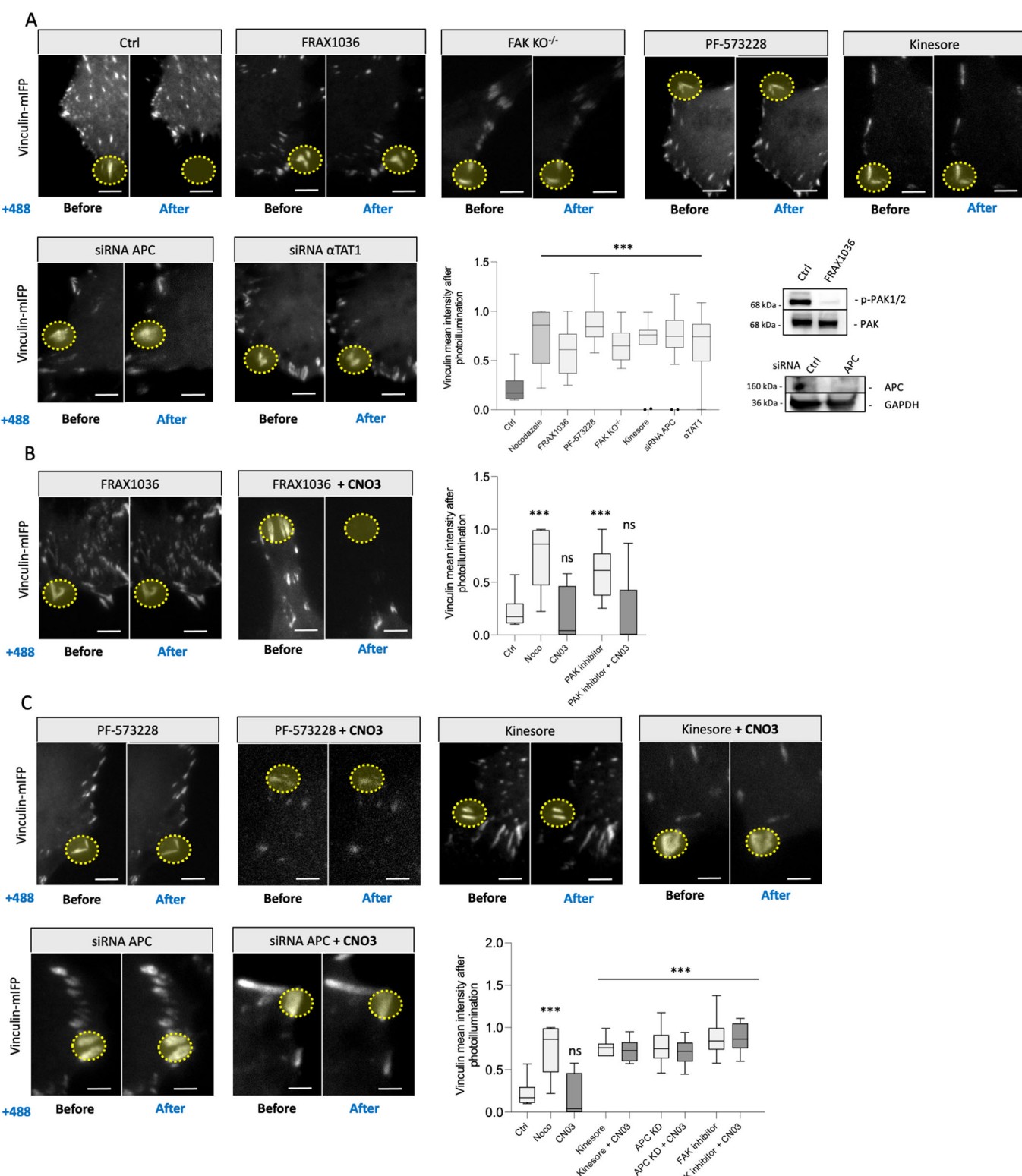

adhesion molecule density gripping the substrate, and (e) density of active GEF-H1 molecules in the cytoplasm proximal to the focal adhesion. The model is based on the following assumptions: (1) growing microtubules arrive to the focal adhesion at a constant rate

and pause there before withdrawal from the adhesion. Based on the comparison of the microtubule dynamics at the cell periphery in control and KANK1/2 depleted cells (Bouchet et al, 2016), it is reasonable to suggest that the average pause is an increasing

Figure 5. OptoKANK mediated disassembly of focal adhesions requires activity of FAK, PAK, Kinesin-1, αTAT1, and APC.

(A) Representative images of Vinculin-mIFP-transfected HT1080 cells carrying the OptoKANK constructs before and after blue light illumination of the focal adhesion (yellow dotted line) for control cells (Ctrl), cells treated with FRAX1036, PF-57328, Kinesore, as well as in APC-depleted, αTAT1-depleted cells, and in FAK$^{-/-}$ knockout MEF cells. Graph shows the normalized vinculin intensity after the illumination of cells treated as indicated (Data are presented as box and whisker plot; Ctrl, $n = 18$ cells; Nocodazole, $n = 12$ cells; FRAX1036, $n = 15$ cells; PF-573228, $n = 10$ cells; FAK KO$^{-/-}$, $n = 8$ cells; Kinesore, $n = 11$ cells; siRNA APC, $n = 18$ cells; αTAT1, $n = 10$ cells; one-way ANOVA, ***$p < 0.001$; the center line denotes the median value (50th percentile) while the box contains the 25th to 75th percentiles of dataset. The whiskers mark the minimum and maximum percentiles; scale bar 5 µm). Immunoblots of phospho-PAK1/2, PAK, APC, and GAPDH are shown in the black box. (B) Representative images of Vinculin-mIFP-transfected HT1080 cells carrying the OptoKANK constructs before and after blue light illumination on the focal adhesion (encircled by line) for cells treated with FRAX1036, and cells treated with FRAX1036 simultaneously with the contractility activator CNO3. Graph shows the normalized vinculin intensity after the illumination for cells under these conditions (Data are presented as box and whisker plot; Ctrl, $n = 18$ cells; Nocodazole (Noco), $n = 12$ cells; PAK inhibitor, $n = 15$ cells, PAK inhibitor + CNO3, $n = 10$ cells; one-way ANOVA, ***$p < 0.001$; the center line denotes the median value (50th percentile) while the box contains the 25th to 75th percentiles of dataset. The whiskers mark the minimum and maximum percentiles; scale bar 5 µm). (C) Representative images of Vinculin-mIFP-transfected HT1080 cells carrying the OptoKANK constructs before and after blue light illumination of the focal adhesion (encircled by line) for cells treated with FAK inhibitor PF-57328, Kinesine-1 modulating drug Kinesore, and APC-depleted cells and such cells treated with Rho activator CNO3. Graph shows the normalized vinculin intensity after the illumination for cells under treatments mentioned above (Data are presented as box and whisker plot; Ctrl, $n = 18$; Nocodazole, CNO3, $n = 8$ cells; Kinesore, $n = 10$ cells; Kinesore + CN03, $n = 16$ cells; APC KD, $n = 18$ cells; APC KD + CNO3, $n = 14$ cells, FAK inhibitor; $n = 10$ cells; FAK inhibitor + CNO3, $n = 10$ cells; one-way ANOVA, ***$p < 0.001$; the center line denotes the median value (50th percentile) while the box contains the 25th to 75th percentiles of dataset. The whiskers mark the minimum and maximum percentiles; scale bar 5 µm).

function of the number of KANK molecules on the focal adhesion. Since the effects of microtubules on focal adhesion depends on kinesin-1 ((Krylyshkina et al, 2002) and this study), microtubules may deliver to focal adhesions some factors which facilitate dissociation of integrins and talins, and because KANK binds to talin, of KANK molecules themselves. Indeed, the fluorescence intensity of talin-1, integrin beta3, and KANK-KN domain started to decrease after 30–60 s of illumination, much earlier than vinculin (Appendix Fig. S5A,B). (2) Based on the known affinity of GEF-H1 to microtubules (Krendel et al, 2002), we assume that any time a microtubule arrives at the focal adhesion, a certain number of GEF-H1 molecules are locally absorbed from the cytoplasm; when this microtubule leaves the focal adhesion, after a pause, GEF-H1 molecules are released back into the cytoplasm, where they diffuse and undergo activation (Azoitei et al, 2019). Rates of myosin activation increases when active GEF-H1 concentration in the cytoplasm is above a threshold. To not overwhelm the model, we do not explicitly include steps of activation of the released GEF-H1 molecules or ROCK-dependent pathway of myosin activation. In agreement with our observations, we assume in our model that activation of GEF-H1 and myosin-II occurs in the centripetal direction from focal adhesion, in the region where the microtubule tips retreated from the focal adhesion are located. (3) We assume that the pulling force applied to the focal adhesion is directly proportional to the myosin density in the model and that the focal adhesion is gripping the substrate if the ratio of the pulling force to adhesive strength is below a threshold, and slipping if this ratio is above the threshold. The slipping of focal adhesion triggers its rapid disassembly (Shemesh et al, 2005) (as manifested by the drop in vinculin concentration). Finally, (4) we assume that the adhesive strength is proportional to the KANK density on the focal adhesion (not that KANK directly contributes to the adhesive strength, but rather that its adhesive molecular partner talin does, and that talin amount is proportional to that of KANK). The feedbacks included into the model are shown in Fig. 6F. Model details, parameters and variations are discussed in the supplemental text.

The predicted time series of the key molecular densities upon local activation of KANK shown in Fig. 6G compare well with the experimental data. The model predicts that up to ~60 s after the illumination, microtubules arriving to the focal adhesion locally sequester GEF-H1 molecules, sharply depleting GEF-H1 density near the focal adhesion, but after that the microtubule detachment leads to release of the accumulated GEF-H1 molecules. If the GEF-H1 dynamics was strictly local, then GEF-H1 molecules would be first captured by the microtubules, then released in the same place, with no net gain. Crucially, due to diffusion, and also because the sink for GEF-H1 on the increasing number of microtubules occurs earlier than the source of GEF-H1 from the decreasing number of microtubules, GEF-H1 molecules initially diffuse closer to the focal adhesion, down the gradient created by the sink. After that, when GEF-H1 is released from microtubules detached from the adhesion, the net GEF-H1 concentration increases after a short delay in the vicinity of the proximal end of the focal adhesion. Effectively, the initial microtubule number increase, counter-intuitively, helps by 'soaking' GEF-H1 into the focal adhesion vicinity, then releasing increased amounts of GEF-H1. Because this diffusion-reaction process introduces delays, GEF-H1 concentration becomes greater than the baseline ~120 s after the local activation of KANK. This leads to a significant additional activation of myosin and traction force increase by ~150 s after the local activation of KANK. By that time, the adhesion is significantly weakened, the force to adhesive strength ratio exceeds the gripping-to-slipping threshold, and the biphasic process comprising of weakening of adhesion to the substrate and local GEF-H1/RhoA/ROCK dependent activation of contractility leads to slippage (Fig. 6G). The model also correctly accounts for the results of several perturbation experiments (Fig. EV1A–D). The model predicts that: (1) Local KANK activation in GEF-H1 KD cell does not result in the traction force increase, hence no adhesion sliding (Fig. EV1A). (2) In cells with Rho activated by CNO3, either with or without functional GEF-H1, there is an elevated pulling on the focal adhesion before the activation of KANK, but the force to adhesion strength ratio is still below the threshold that causes sliding. After the activation of KANK, microtubules' arrival brings adhesion weakening, which, together with the elevated pulling, triggers the sliding (Fig. EV1B,C). (3) When the ROCK activity is weakened by Y27632, despite the Rho activity increase due to the GEF-H1-dependent activation, myosin-II-dependent pulling force is too weak to slide the adhesion even against the weakened adhesion (Fig. EV1D).

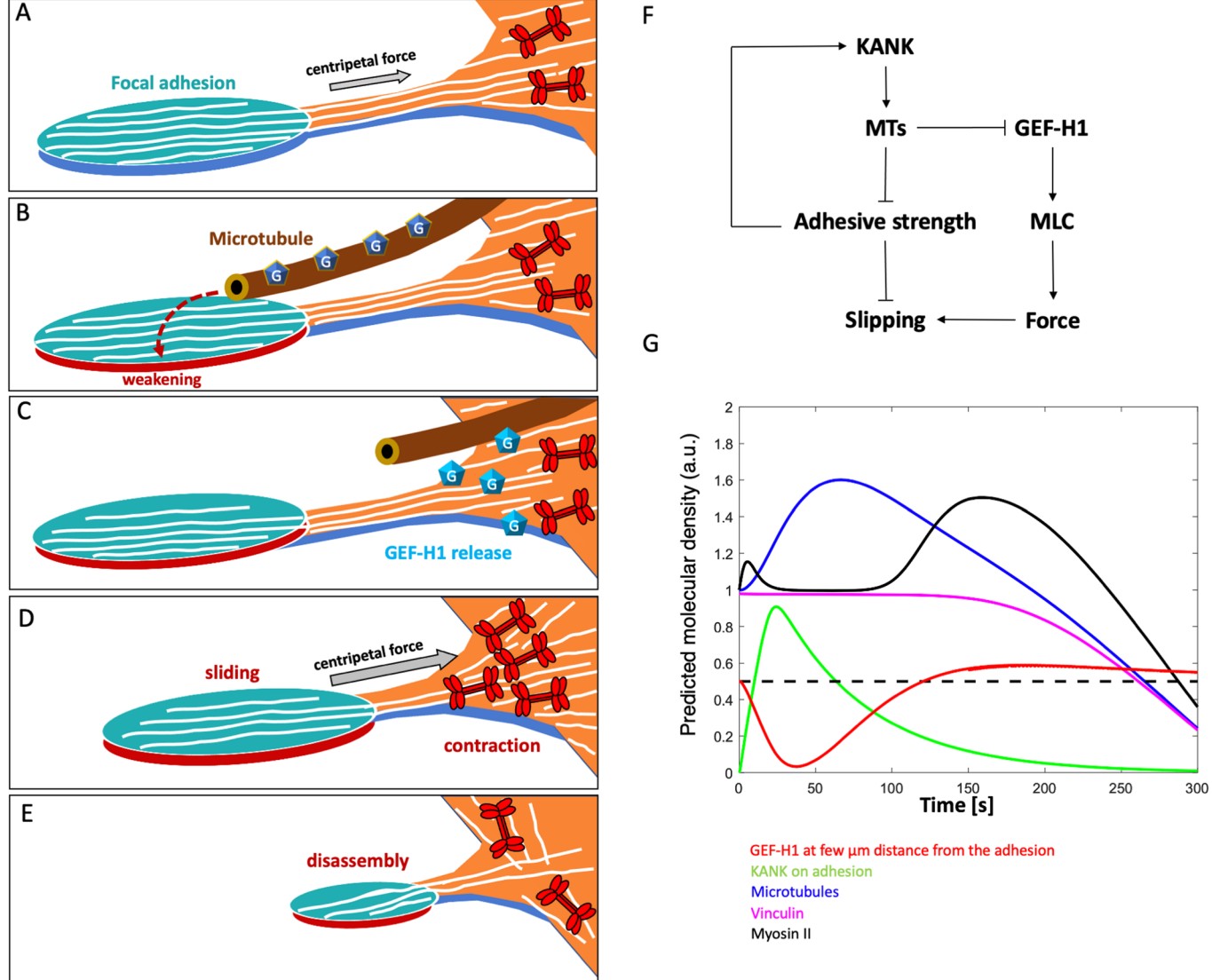

**Figure 6. Working hypothesis and a physical model of the process of microtubule-driven focal adhesion disassembly.**

(A–D) Cartoons depicting the proposed model of sequential events leading to the microtubule-driven focal adhesion disassembly. (A) Focal adhesion and associated stress fiber embedded in contractile actomyosin network are depicted. Actin filaments are schematically presented by white lines and myosin II filaments by red symbols (not in scale). The focal adhesion is stabilized by centripetal force generated by the actomyosin network. (B) Microtubule attachment to focal adhesion weakens it but the centripetal force is not yet sufficient to promote the sliding. Microtubule is associated with non-active GEF-H1 molecules (dark blue pentagons with "G" inside). (C) Microtubule detachment from the focal adhesion triggers the release of GEF-H1 and its activation in the vicinity of the proximal end of the focal adhesion (represented by changes in the color of the symbols to light blue). (D) Active GEF-H1 triggers the Rho/ROCK pathway resulting in formation of new myosin and actin filaments and local actomyosin contraction. The strong centripetal force developed as a result of such contraction leads to sliding of weakened focal adhesion. (E) Sliding of the focal adhesion leads to it disassembly. (F) Diagram showing the feedbacks included into the physical model. (G) Model-predicted time series for dynamics of microtubules, vinculin, KANK, GEF-H1, and myosin-II upon optogenetic induction of microtubules contact with the focal adhesion (see text). Model parameters used in the simulations are described in the Appendix. Dashed line indicates the threshold above which GEF H1 significantly affects the myosin activation pathway. See also EV1.

## Discussion

In this study, we used a novel optogenetic method to decipher the process of microtubule-driven focal adhesion disassembly. We have shown that transient attachment of microtubule tips to a focal adhesion followed by their detachment results in a local burst of actomyosin contractility, which is a critically important step in the process that triggers focal adhesion sliding and consequently its disassembly.

The optogenetic construct promoting association of microtubule tips with focal adhesions consisted of two halves of KANK1 protein, talin binding domain (KN) and the rest of the molecule (ΔKN) that can be linked by illumination. The KN-containing construct (mApple-KN-LOV2ssrA) localizes to focal adhesions and likely displaces the endogenous KANK proteins from them (cf Rafiq et al, 2019) thereby uncoupling focal adhesions and microtubules. This scenario is supported by the fact that the mean focal adhesion area in cells expressing OptoKANK constructs was

larger than that in mock-transfected cells, similarly to the situation in cells with depolymerized microtubules or KANK1 knockdown (Rafiq et al, 2019). We have further shown that local illumination of focal adhesions containing mApple-KN-LOV2ssrA resulted in accumulation of ΔKN containing construct (SspB-ΔKN-mEmerald) in these focal adhesions apparently restoring the integrity of the KANK1 molecule. This in turn, resulted in a transient increase of microtubule tips overlapping with focal adhesions and augmentation of the delivery of membrane to the focal adhesion area. Remarkably, optogenetic targeting of microtubules to peripheral adhesions leads to the disassembly of these adhesions similarly to that observed in cells recovering following washout of microtubule-disrupting drugs (Ezratty et al, 2005). Thus, our optogenetic approach permitted us to investigate in detail the kinetics of microtubule-driven focal adhesion disassembly and identify the major molecular players participating in this process.

The key observation was that the initial increase of the number of microtubules associated with focal adhesion was followed by withdrawal of microtubules from the focal adhesion. Mechanism of this withdrawal is not completely clear. Microtubule targeting to the focal adhesions resulted in some decrease of amount of integrin and talin earlier than decrease of vinculin. Reduction of talin brings about reduction of KANK which in turn should reduce association of microtubules with the adhesion. Of note also that contact with focal adhesion can trigger the microtubule shrinking (catastrophes) (Efimov and Kaverina, 2009).

Withdrawal of the microtubules from focal adhesion zone is followed by sliding and gradual disassembly of the focal adhesion. Our previous studies suggested that detachment of microtubules from integrin adhesions triggers the release of GEF-H1 from microtubules and subsequent formation of myosin-II filaments (Rafiq et al, 2019). In agreement with this idea, we have shown here that (i) GEF-H1 is critically important for triggering microtubule-driven focal adhesion sliding and disassembly; (ii) local assembly of myosin-II and actin filaments is observed near the proximal end of focal adhesion after microtubule retraction; (iii) inhibition of actomyosin contractility by Y27632 prevents the sliding and disassembly of illuminated focal adhesions; (iv) activation of Rho by CNO3 restores the disassembly of focal adhesion upon illumination in GEF-H1 knockdown cells. Altogether, these data suggest that local retraction of microtubules after their accumulation above a focal adhesion upon OptoKANK activation results in release and activation of GEF-H1, which in turn triggers activation of Rho. This activation leads to local formation of an actomyosin contractile network in proximity to the focal adhesion, and development of traction forces pulling the adhesion in a centripetal direction. Furthermore, we detected the increase of traction force at the focal adhesion area that occurs after microtubule retraction and showed that it preceded the sliding and disassembly of focal adhesion.

In our model, the focal adhesion disassembly occurs as a result of focal adhesion sliding. Indeed, focal adhesions are mechanosensitive in a sense that they undergo rapid disassembly in the absence of stretching force generated by actomyosin contractility (Balaban et al, 2001; Chrzanowska-Wodnicka and Burridge, 1996; Even-Ram et al, 2007; Helfman et al, 1999; Riveline et al, 2001; Vicente-Manzanares et al, 2007). Detached focal adhesion sliding with low frictions should not experience any significant stretching force and therefore should undergo rapid disassembly similar to

that induced by inhibition of myosin contractility (Shemesh et al, 2005). Obviously, there could be situations when sliding of focal adhesions is not accompanied by their disassembly, most probably because the friction forces experienced by the adhesion could in some cases be sufficient to keep them stretched during sliding. Our measurements detected increase of traction forces in proximity of illuminated focal adhesion before sliding, but technical limitation of the system did not allow to directly measure the traction force during the sliding. We assume that OptoKANK activation and microtubule targeting to focal adhesions somewhat weaken the adhesion strength, making the adhesions prone to detachment by a transient local increase in actomyosin contractile forces. This assumption is in agreement with decrease of amount of integrin β3, which we detected already during the first minute after onset of illumination.

To elucidate the process of microtubule-driven focal adhesion disassembly we conducted a screen revealing the involvement of key molecular players in this process. In the course of these studies, we did not find evidence of involvement specific proteolysis or endocytosis in the process of microtubule-driven focal adhesion disassembly. Indeed, inhibitors of calpain, matrix metalloproteases (MMP) and dynamin, did not interfere with focal adhesion disassembly induced by OptoKANK activation. Several other molecular players suggested in the literature were also not confirmed in our experimental system. In part, this discrepancy could be attributed to the difference between experimental systems: local targeting of microtubules to focal adhesions used in our study versus global microtubule outgrowth after nocodazole washout used in majority of previous studies.

We identified several proteins, knockdown or/and inhibition of which efficiently prevented the microtubule-driven focal adhesion disassembly after OptoKANK activation. Besides the Rho activation axis (GEF-H1, ROCK) we found that kinases FAK and PAK, microtubule motor kinesin-1, tubulin acetylase αTAT1, and microtubule-associated activator of actin polymerization APC, are needed for the focal adhesion disassembly. The involvement of FAK, PAK, and kinesin-1 is consistent with the previous publications (Ezratty et al, 2005; Kiosses et al, 1999; Krylyshkina et al, 2002).

To clarify whether FAK, APC, PAK, and Kinesin-1 participate in local microtubule-driven activation of Rho and actomyosin contractility similar to GEF-H1, we investigated whether pharmacological activation of Rho during illumination of focal adhesions can overcome the effects of depletion or inhibition of these proteins. While activation of Rho rescued the effects of inhibition of PAK and GEF-H1, it did not rescue the effects of FAK, APC, and kinesin-1 inhibition. Previous studies suggest that the involvement of FAK in the focal adhesion disassembly (Hamadi et al, 2005) could be mediated by activation of Src and p190RhoGAP (Schober et al, 2007; Wu et al, 2016) leading to inhibition of Rho and myosin contractility. We found that activation of Rho by CNO3 did not interfere with microtubule-driven focal adhesion disassembly suggesting that FAK may also work downstream of Rho. Also, in our system, FAK can hardly promote focal adhesion disassembly through interactions with dynamin (Ezratty et al, 2005) because inhibition of dynamin by dynasore and hydroxydynasore, which inhibit endocytosis, was not sufficient to prevent the effect of OptoKANK activation. Thus, further studies are needed to elucidate the function of FAK in the microtubule-driven focal

adhesion disassembly. Similarly, the role of APC and αTAT1 requires clarification. A plausible function of kinesin-1 is a transportation along microtubules of the factors weakening the focal adhesions. Of note, the identified players can work synergistically, as kinesin-1 is known to be involved in the transport of APC along microtubules (Juanes et al, 2017; Ruane et al, 2016), while microtubule acetylation by αTAT1 can affect association of GEF-H1 with microtubules (Seetharaman et al, 2022).

While molecular details of microtubule-driven focal adhesion disassembly remain to be elucidated, our study provided strong evidence in favor of a new model of this process. We have shown that the key event induced by microtubule targeting to focal adhesion is local GEF-H1-dependent activation of myosin filament formation and actomyosin contractility near the proximal end of the focal adhesion. This local contraction triggers the focal adhesion sliding, which in turn may result in its disassembly. Future studies will clarify how this mechanism functions during 2D and 3D cell migration and determine if this mechanism can be applied to the interaction of microtubules with other types of integrin adhesions like podosomes and fibrillar adhesions.

# Methods

## Cell culture and cell transfection procedures

The HT1080 human fibrosarcoma cell line was obtained from the American Type Culture Collection and cultured in MEM supplemented with 10% heat-inactivated FBS, non-essential amino acids and sodium pyruvate (Sigma-Aldrich), in an incubator at 37 °C and 5% $CO_2$. FAK$^{-/-}$ MEF cells were a gift from P. Kanchanawong (Mechanobiology Institute, Singapore) and were cultured in DMEM supplemented with 10% heat-inactivated FBS, non-essential amino acids and sodium pyruvate (Sigma-Aldrich) and penicillin–streptomycin (Thermo Fisher Scientific), in an incubator at 37 °C and 5% $CO_2$.

Cells were transiently transfected with the expression vector plasmids using electroporation (Neon Transfection System, Life Technologies) in accordance with the manufacturer's instructions. Specifically, one pulse of 950 V of 50 ms was used for HT1080 cells, one pulse of 1350 V of 20 ms was used for the FAK$^{-/-}$ MEF cells. For siRNA-mediated knockdown, HT1080 cells were transfected at the following concentrations: 100 nM for GEF-H1 (Dharmacon, ON-TARGETplus siRNA, cat. no. J-009883-09-0002), 100 nM for αTAT1 (ThermoFisher siRNA cat. no. AM16708), 100 nM for APC (Dharmacon, cat. no. L-003869-00-0005), 100 nM for BNIP2, and 100 nM for ARP2. For control experiments, cells were transfected with non-targeting pool siRNA (ON-TARGETplus, Dharmacon cat. no. D-001810-10) at a concentration similar to that of the gene-targeted siRNAs. Cells were transfected using Lipofectamine RNAiMAX (Invitrogen) according to the manufacturer's instructions.

## Plasmids

For generation of iLID-based optogenetic constructs of KANK (OptoKANK), the KN domain of KANK1 corresponding to amino acid residues 1–68 was fused with the mApple fluorescence tag and

LOV2ssrrA domain (mApple–KANK-KN–LOV2ssrA) (Guntas et al, 2015). The rest of KANK1, ΔKN, corresponding to amino acid residues 69–1352, was fused with the mEmerald fluorescence tag and the SspB domain (SspB–ΔKN–mEmerald) (Guntas et al, 2015). The mApple–KN–LOV2ssrA and SspB–ΔKN–mEmerald constructs were cloned by Epoch Life Science. For generation of ITGB3-mIFP and mIFP-Talin1 constructs, the GFP fluorescent tags of ITGB3-GFP (gift from Jonathan Jones - Addgene plasmid #26653) and GFP-Talin1 (gift from Anna Huttenlocher - Addgene plasmid #26724) were swapped out with the mIFP fluorescent tag of the mIFP-Vinculin (kindly provided by Dr. Michael W. Davidson, Florida State University, FL, USA). Constructs were cloned by Epoch Life Science.

## Live cell observation

Pharmacological treatments were performed using the following concentrations of inhibitors or activators: 1 μM for Nocodazole (Sigma-Aldrich), 0.4 μM for Y-27632 dihydrochloride (Sigma-Aldrich), 20 mM for Acetyl-Calpastatin (Tocris, Cat. No. 2950), Monastrol (Merk, cat. No. 254753-54-3), 80 μM for Dynasore (Abcam Cat. No. 120192), Ilomastat (GM6001 Tocris Cat. No. 2983), Tubacin (Cat. No. 3402), MAP4K4-IN-3 (MedChemExpress, Cat. No.: HY-125012), 1 μM for FRAX1036 (MedChemExpress Cat. No. HY-19538), 50 μM for Kinesore (Cat. No. 6664), 5 μM for PF-57328 (Tocris, Cat. No. 3239), and 1 μg ml$^{-1}$ for Rho activator II (CNO3, Cytoskeleton). Cells were treated at least 2 h at 37 °C and 5% $CO_2$, except when specified in the legend, before OptoKANK-based live imaging. For nocodazole-washout experiments, transfected HT1080 cells were plated on 35-mm dish from IBIDI overnight. One hour before imaging, nocodazole in fresh MEM medium with 10% FBS was added to the cells. The 35-mm dishes were mounted in a perfusion chamber (CM- B25-1, Chamlide CMB chamber). Nocodazole was washed out using fresh MEM medium with FBS 2 h later. For the contractility rescued experiments, cells were first depleted 24 h or treated with inhibitors for 2 h before addition of CNO3 at 1 μg mg.ml$^{-1}$. Cells were then imaged using microscope 1 h after addition of CNO3 without changing the medium.

## Fluorescence microscopy

OptoKANK-based live experiments were performed using total internal reflection fluorescence (TIRF) microscopy (Olympus IX81, Zero Drift Focus Compensator, Dual camera Hamamatsu ORCA-Fusion BT, Objective ×100) or using confocal microscopy (Yokogawa CSU-W1, Nikon TiE, 2x Photometrics Prime 95b CMOS camera, Objective ×63) using Metamorph software. For OptoKANK activation, the 488 nm wavelength laser was set up at 1% in Fluorescence Loss in Photobleaching (FLIP) mode on the region of interested (Intensity FRAP power = 0.16 μW/μm$^2$). Classical time-course experiments were performed at 5 s intervals with continuous photoactivation (except every 5 s corresponding to the acquisition steps), and 3 s intervals for EB1 experiments.

## Immunoblotting

Cells were lysed directly in Laemmli buffer for Western blot (Tris–HCl pH 6.8 0.12 M, glycerol 10%, sodium dodecyl sulfate 5%,

β-mercaptoethanol 2.5%, bromophenol blue 0.005%) and extracted proteins were separated by SDS–PAGE in 4–20% SDS–polyacrylamide gel (Thermo Fisher Scientific) and transferred to polyvinylidenedifluoride membranes (Bio-Rad) at 75 V for 2 h.

Subsequently, the polyvinylidenedifluoride membranes were blocked for 1 h with 5% bovine serum albumin (BSA, Sigma-Aldrich), then incubated overnight at 4 °C with appropriate antibodies: anti-GEF-H1 (Cell Signaling, cat. no. 4145, dilution 1:1000); anti-α-tubulin (Sigma-Aldrich, cat. no. T6199, dilution 1:3000); anti-GAPDH (Santa Cruz Biotechnology, cat. no. sc-32233, dilution 1:3000); anti-APC (Thermo Fisher Scientific, cat. no. A5-35188, dilution 1:1000); anti-ARP2 (Cell Signaling, cat. no. #3128, dilution 1:1000); anti-Phospho-PAK1 (Ser199/204)/PAK2 (Ser192/197) (Cell Signaling, cat. no. #2605, dilution 1:2000); anti-acetyl-α-Tubulin (Lys40) (Cell Signaling, cat. no. #3971, 1:1000); anti-αTAT1 (Thermo Fisher Scientific, cat. no. PA5-112992, dilution 1:1000); anti-cleaved Caspase-3 (Cell Signaling Technology, cat. no. #9661, 1:1000); anti-mApple antibody (St. John's Laboratory cat. no. STJ140269, 1:1000), anti-mEmerald antibody (St. John's Laboratory cat. no. STJ140226, 1:1000), anti-Phospho-FAK (Y397) (Cell Signaling, cat. no. #3283, 1:1000).

The membranes were washed three times (10 min each) and probed by incubation for 1 h with the appropriate secondary antibodies conjugated with horseradish peroxidase (Bio-Rad). The membranes were then washed three times (15 min each at room temperature), developed using PierceTM ECL western blotting substratum (Thermo Fisher Scientific) and imaged using a ChemiDoc imaging system (Bio-Rad).

## Micropatterning of adhesive islands

Fibronectin-patterned glass coverslips were microfabricated using the first steps of the glass technique described by Vignaud et al (Vignaud et al, 2014). Briefly, glass coverslips (VWR) were plasma treated for 30 s and incubated for 30 min at room temperature with 0.1 mg.ml$^{-1}$ poly-L-lysine-grafted-polyethylene glycol (pLL-PEG, SuSoS) diluted in HEPES (10 mM, pH 7.4, Sigma). After washing in deionized phosphate-buffered saline (dPBS, Life Technologies), the pLL-PEG covered coverslip was placed with the polymer brush facing downwards onto the chrome side of a quartz photomask (Toppan) for photolithography treatment (5 min UV-light exposure, UVO Cleaner Jelight). Subsequently, the coverslip was removed from the mask and coated with 30 μg.ml$^{-1}$ fibronectin (Sigma) diluted in dPBS for 30 min at RT.

## EB3 comet tracking and data analysis

Images of EB3-mIFP acquired from the TIRF microscope were exported as multidimensional TIFF files. EB3 comets from these raw unprocessed images were tracked automatically using the plusTipTracker software (Applegate et al, 2011; Matov et al, 2010). To measure displacement, lifetime and velocities of EB3 comets, the following parameters were set in the program: search radius range, 4–15 pixel; minimum subtrack length, 3 frames; maximum gap length, 10 frames; maximum shrinkage factor, 0.8; maximum angle forward, 50; maximum angle backward, 10; and fluctuation radius, 2.5. To visualize comet tracks in individual growth cones, the plusTipSeeTracks function was used. MT dynamics parameters were compiled from multiple individual experiments.

## Traction force microscopy

PDMS micropillars were fabricated to form PDMS mold for micropillar array. After silanizing the surface of the PDMS pillars with Trichloro (1H,1H,2H,2H-perfluorooctyl) silane (Sigma, 448,931) overnight, new PDMS (DOWSIL 184 silicone elastomer, Dow Corning, MI, USA) was directly cast onto the surface of the micropillar to make a PDMS mold with holes. After degassing for 15 min, the mold was cured at 80 degrees for 2 h. The PDMS mold was peeled off from the PDMS pillars, cut 1 cm square and placed on plastic dishes face up following a silanization of their surface with Trichloro (1H,1H,2H,2H-perfluorooctyl) silane overnight.

To fabricate the array of micropillar, whose refraction index is similar to that of the growth medium, a small drop of My-134 polymer (My Polymers Ltd., Israel) was put on the center of coverslip coated with 3-(Trimethoxyilyl)propyl methacrylate (sigma, 440,159) and then, the silanized PDMS mold covered the drop face down onto the coverslip with thin layer of My-134 polymer for 15–30 min. After degassing for 5–15 min to get rid of air bubbles inside the polymer, the assembly was placed in a cell culture dish, covered with fresh milli-Q water and cured under short wavelength UV radiation (UVO Cleaner 342A-220, Jelight Company Inc., USA) for 6 min. Then, the PDMS mold was carefully peeled off from the coverslip.

Top of My-134 pillars were coated with fluorescence-labeled fibronectin. Briefly, PDMS stamps were incubated with solution containing 30 μg/ml fibronectin and 2 μg/ml fibrinogen Alexa Fluor 647 conjugate solution (F35200, Invitrogen, USA) in Dulbecco's phosphate-buffered saline (Sigma- Aldrich) at room temperature for 90 min. After washing with Milli-Q water and air-drying the surface, the PDMS stamp was put onto the top of My-134 pillars freshly exposed to UV-Ozone (UV Ozone ProCleaner Plus, BioForce Nanosciences). After 5 min of contact, the stamp was removed. Before cell plating, the My-134 pillars were incubated with 0.2% Pluronic F-127 (Sigma) for 1 h for blocking, followed by washing three times with Dulbecco's phosphate-buffered saline. The pillars in the array were arranged in a triangular lattice with 4 μm center-center distance and the dimensions of pillars were d = 2.1 μm with h = 6 μm (k = 50 nN/μm). The traction forces by fluorescent-labeled My-134 pillars were calculated using a custom-build MATLAB program (version 2019a, MathWorks) as described previously (Doss et al, 2020).

## Immunofluorescence microscopy

Cells were fixed with 3.7% PFA (Sigma) for 20 min, permeabilized with 0.1% Triton in PBS (Sigma), then washed with PBS, and blocked with a blocking solution (2.5% bovine serum albumin in PBS Tween 0.2%) for 1 h. Samples were incubated overnight at 4 °C with primary antibody in blocking solution: anti-vinculin (Sigma-Aldrich, catalogue no. V9131, dilution 1:400) followed by three washes with PBS Tween 0.2%. The cells were then incubated with secondary antibody at room temperature for 1 h followed by three washes with PBS Tween 0.2%.

## Physical model

See Appendix.

## Statistical analyses

Statistical analyses were performed using GraphPad Prism software (GraphPad, version 9). Statistical significance was determined by the specific tests indicated in the corresponding figure legends.

# Data availability

https://www.ebi.ac.uk/biostudies/bioimages/studies/S-BIAD1039.

The source data of this paper are collected in the following database record: biostudies:S-SCDT-10_1038-S44318-024-00114-4.

# Peer review information

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

## Acknowledgements

We thank Christophe Guilluy for support and encouragements, A. Wong (MBI, Singapore) for expert help in paper editing, Bryant L. Doss for advices with the TFM experiments, Yukako Nishimura (Hokkaido University) for advices with the MY-134 pillars technology, M. Davidson fluorescence protein collection (The Florida State University, Tallahassee, USA), the SIMBA microscopy facility, the Wet Lab and nanofabrication core facility at the Mechanobiology Institute for technical help. The research is supported in part by the National Research Foundation, Prime Minister's Office, Singapore, Singapore Ministry of Education under the Research Centres of Excellence program through the Mechanobiology Institute at the National University of Singapore (refs no. R-714-006-006-271, A-0003467-01-00 and A-0003467-00-00), by the National Research Foundation Singapore under its Mid-Sized Grant (NRF-MSG-2023-0001-0003), by the Singapore Ministry of Education Academic Research Fund Tier 2 (MOE Grant No: MOE2018-T2-2-138, awarded to ADB; MOE2019-T2-1-099 and MOE2019-T2-02-014; awarded to PK), and Tier 3 (MOE Grant No: MOE2016-T3-1-002 and MOET32021-0003; awarded to BC and ADB), as well as by Neurodis Foundation (awarded to JA, ref R2311CC), and by US National Science Foundation (DMS1953430, awarded to AM).

## Author contributions

**Julien Aureille**: Conceptualization; Resources; Data curation; Software; Formal analysis; Funding acquisition; Validation; Investigation; Visualization; Methodology; Writing—original draft; Writing—review and editing. **Srinivas S Prabhu**: Resources; Investigation. **Sam F Barnett**: Resources. **Aaron J Farrugia**: Investigation. **Isabelle Arnal**: Resources. **Laurence Lafanechère**: Resources. **Boon Chuan Low**: Resources. **Pakorn Kanchanawong**: Resources. **Alex**

**Mogilner**: Validation; Investigation; Visualization; Methodology; Writing—original draft; Writing—review and editing. **Alexander D Bershadsky**: Conceptualization; Resources; Data curation; Software; Formal analysis; Supervision; Funding acquisition; Validation; Investigation; Visualization; Methodology; Writing—original draft; Project administration; Writing—review and editing.

Source data underlying figure panels in this paper may have individual authorship assigned. Where available, figure panel/source data authorship is listed in the following database record: biostudies:S-SCDT-10_1038-S44318-024-00114-4.

## Disclosure and competing interests statement

The authors declare no competing interests.

# Expanded View Figure

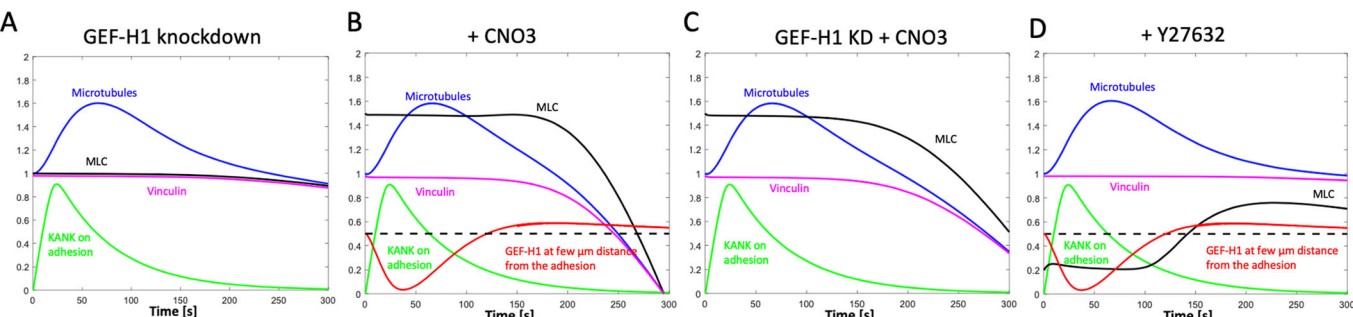

**Figure EV1.** **Model-predicted time series for microtubule, vinculin, KANK, GEF-H1 and myosin densities upon simulated perturbations.**

The following perturbations: GEF-H1 knockdown (**A**), treatment with CNO3 (**B**), GEF-H1 knockdown + CNO3 (**C**) and treatment with Y27632 (**D**) have been simulated based on model parameters described in the Appendix. Dashed line indicates the threshold above which GEF H1 significantly affects the myosin activation pathway.

