## [Peer Review File · The EMBO Journal]

Focal adhesions are controlled by microtubules through local contractility regulation

Julien Aureille, Srinivas Prabhu, Sam Barnett, Aaron Farrugia, Isabelle Arnal, Laurence Lafanechère, Boon Chuan Low, Kanchanawong Pakorn, Alex Mogilner, and Alexander Bershadsky

Corresponding authors: Julien Aureille (julien.aureille@gmail.com) , Alexander Bershadsky (mbiba@nus.edu.sg)

Review Timeline:

Submission Date:	20th Aug 23
Editorial Decision:	27th Sep 23
Revision Received:	13th Feb 24
Editorial Decision:	8th Mar 24
Revision Received:	12th Apr 24
Accepted:	15th Apr 24

Editor: Ieva Gailite

Transaction Report:

Dear Dr. Aureille,

Thank you for submitting your manuscript for consideration by the EMBO Journal. We have now received comments from three reviewers, which are included below for your information.

As you will see from the reports, all reviewers find the study of interest, while also pointing out a number of important aspects that would need to be strengthened before they can recommend acceptance of the manuscript. In particular, referees #1 and #2 highlight that a better characterisation of the effects of KANK1 overexpression would be needed. Additionally, reviewer #3 asks for better integration of the proposed mechanism of focal adhesion disassembly in the context of the focal adhesion maturation-disassembly cycle.

Based on the interest expressed in the reports, I would like to invite you to address the issues raised by the referees in a revised manuscript. I think it would be useful to discuss the revision in more detail via email or phone/videoconferencing - please let me know which option you prefer.

We generally allow three months as standard revision time. As a matter of policy, competing manuscripts published during this period will not negatively impact on our assessment of the conceptual advance presented by your study. However, please contact me as soon as possible upon publication of any related work to discuss the appropriate course of action. Should you foresee a problem in meeting this deadline, please let us know in advance to discuss an extension.

When preparing your letter of response to the referees' comments, please bear in mind that this will form part of the Review Process File and will therefore be available online to the community. For more details on our Transparent Editorial Process, please visit our website: <https://www.embopress.org/page/journal/14602075/authorguide#transparentprocess>. Please also see the attached instructions for further guidelines on preparation of the revised manuscript.

Please feel free to contact me if you have any further questions regarding the revision. Thank you for the opportunity to consider your work for publication. I look forward to discussing your revision.

With best regards,

Ieva

- a point-by-point response to the referees' comments, with a detailed description of the changes made (as a word file).
- a word file of the manuscript text.
- individual production quality figure files (one file per figure)

- a complete author checklist, which you can download from our author guidelines (<https://www.embopress.org/page/journal/14602075/authorguide>).
- Expanded View files (replacing Supplementary Information)
Please see out instructions to authors
<https://www.embopress.org/page/journal/14602075/authorguide#expandedview>

We realize that it is difficult to revise to a specific deadline. In the interest of protecting the conceptual advance provided by the work, we recommend a revision within 3 months (26th Dec 2023). Please discuss the revision progress ahead of this time with the editor if you require more time to complete the revisions.

Referee #1:

The manuscript by Aureille and colleagues is an extension of a previously published paper from the Bershinsky lab (reference 17). The authors state on page 12 that 'The key observation was that the initial increase of the number of microtubules associated with focal adhesion was followed by withdrawal of microtubules from the focal adhesion zone which in turn resulted in sliding and gradual disassembly of the focal adhesion.' I agree with this conclusion. However, the authors should include more control experiments to ensure that their conclusion is correct. (1) HT1080 cells express high levels of KANK2 and lower levels of KANK1, both of which have not been depleted (or even better Crisp/Cas9-deleted) prior to the overexpression of the KANK1 transgene. If this experiment is not carried out it is difficult to exclude that the reported biology is simply due to overexpression of KANK1. (2) The authors fail to demonstrate to what extent they have overexpressed KANK1. I cannot find a quantification of the overexpressed KANK1 (and the endogenous KANK1 and LKANK2). (3) Furthermore, overexpression of KANKs over a certain threshold leads to cell death. It is possible that this does not happen with the optogenetically regulated fusion of two parts of KANK1 into one, and not in HT1080 cells. However, it should be excluded. (3) KANKs can homo- and heterodimerize. Hence, the transgene can form heterodimers with the endogenous KANK1 and KANK2. Whether this happens and how this possibility affects the transgene before and after illumination is not addressed by the authors. (4) Finally, the authors claim that KANK links Talin with the CMSC. However this has never been shown. It would be fantastic if they could show that a single photogenetically fused KANK1 molecular binds talin on the N-terminal side and Liprin/Kif on the C-terminal side. This would strengthen their claim that KANK links FAs and MTs and would exclude that KANK is simply present in two different locations that are accidentally in proximity to each other.

It is also interesting to note that the authors discuss our paper (reference 18) and conclude that the KANK2 mechanism described in reference 18 does not help explaining the findings of their manuscript. Apparently, it even contradicts their findings. We used fibroblasts that were depleted of endogenous KANK2, plated the fibroblasts on FN only and analyzed FA belt slide, which precedes FA disassembly. We did not analyze in reference 18 the role of KANK in FA disassembly. I also have difficulties to understand why a failure to 'reduce podosome numbers upon KANK1 overexpression (is) inconsistent with the idea that KANK2 mediates weakening of talin (ABS2)-actin interaction'? Podosomes express KANK1 and low levels of KANK2, at least in vivo (<https://doi.org/10.1016/j.yexcr.2020.112391>). The claim of the KANK1 podosome connection is again based on overexpression and has little to do with FA belt sliding. I think Bershinsky and colleagues should read reference 18 more carefully and take the different experimental design and aims between the papers into account when they compare results and outcome.

Reinhard Fässler

Referee #2:

This manuscript describes the generation and application of an optogenetic probe that enables modulating the activation of the microtubule binding protein KANK1 with high spatio-temporal control in cells. KANK1 has been previously shown to regulate integrin-mediated cell adhesion through its interaction with talin, an effect that is sensitive to the presence of the RhoA activator GEF-H1. However, the molecular details underlying this phenomenon are not fully understood. In this study, the author established a system to study KANK function using optogenetics. For this, the KANK protein is split in

two parts: The first part is fused to LOV2SsrA, while the second part is attached to SSpB. Since LOV2ssrA and SSpB undergo dimerization upon light illumination at 488 nm, the KANK-mediated activities at focal adhesions (FAs) can be tightly controlled and studied using live cell imaging. The results demonstrate that KANK1-mediated recruitment of microtubules leads to a local increase in myosin-II activity, which requires GEF-H1 and the presence of FAK, PAK, Kinesin-1 and α -TAT. Computer simulations support a model in which microtubule detachment from FAs induces GEF-H1 release triggering myosin activation and FA sliding ultimately leading to FA disassembly.

Overall, the study is very carefully conducted, the data are of high quality and the manuscript is well written. The described effects seem to be consistent with previously proposed mechanisms and the identification of new modulators advance our understanding of microtubule/KANK-mediated control of FA turnover. I have a few minor comments but, in general, recommend the publication of this study.

1. The here presented experiments use an overexpression system, where KANK constructs are expressed on top of the endogenously expressed KANK proteins. I feel that the study would benefit from characterizing the overexpression effects in more detail. For instance, the authors acknowledge in the discussion that FAs in cells expressing the OptoKANK constructs are "significantly larger" than control cells. It would be helpful if this increase in FA size as well as the degree of KANK overexpression were quantified. Which KANK isoforms are endogenously expressed in HT1080 cells? It would be also worth clarifying whether the overall morphology and structure of the actomyosin network are altered compared to the control situation, and whether the degree of FAK activation (pY397 or pY567/577) is affected by the OptoKANK expression. While these experiments will not change the overall outcome of the study, it will help the reader to interpret the experiments.
2. The experiments seem to focus on one type of adhesion structure, namely comparably large, peripheral FAs at the cell edge. An exciting aspect of here presented technique is that distinct FAs classes can be studied. Fig. 2A or Fig. 3A, for instance, show that KANK-KN is also found in centrally located adhesion structures. Does an illumination at these sites also lead to FA sliding and disassembly, as observed for peripheral FAs? Expanding the experiment in this way could increase the impact of the study.
3. All images lack scale bars, please include them. Western blots should indicate molecular weights in main figures and supplementary figure 6. To allow replication of the data on other microscope setups, please indicate the laser intensity that was used for optogenetic modulation in mW/area.
4. It is often unclear, which statistical test was used to evaluate experimental results. In contrast to the statement in "Statistical analyses", I could not find this information in the figure legends of Fig. 3E, Fig. 4A, C, Fig. 5A-C.
5. The technological approach is conceptually highly similar to a previous study, in which light modulation of the LOV2SsrA-SspB dimer is used to study the mechanics of talin-1 (Yu et al. Phys Rev X, 2020). I could not find the reference to this study and recommend to include it.

Referee #3:

Summary:

In this study the authors investigate how microtubule attachment to focal adhesions (FAs) via KANK affects focal adhesion dynamics. Employing a photoinducible system they are able to attract microtubule tips to focal adhesions and then use this system to study how this triggers a sequence of events eventually leading to focal adhesion disassembly. Based on their study they propose a model where microtubule dissociation from focal adhesions leads to release of GEF-H1, activation of the Rho-ROCK-myosin II cascade, increased tension, sliding and finally focal adhesion disassembly. The results are solid and provide important new insights into the role of microtubules in focal adhesion dynamics and the mechanisms involved. The data and novel mechanistic insights are very valuable, however, some of the interpretations need to be reevaluated and placed into the context of the regular FA life cycle.

Main points:

- 1) The authors present microtubule attachment as a mechanism of focal adhesion disassembly and microtubules have generally been ascribed to this role. However, the results presented in this work demonstrate the occurrence of a number of events prior to FA disassembly. Since microtubule attraction is studied in this work only for already established FAs, it is difficult to place these events along the progression of FA maturation and disassembly. It is therefore unclear whether the increase in tension following microtubule disassembly is indeed a specific signal for FA disassembly or could it be part of the regular FA maturation and traction build-up required for motility? To answer this question, it would be important to know at what stage of the FA life cycle microtubules attach to FAs. Does this only occur in late stage FAs?
- 2) One step in the proposed sequential model is not mechanistically explained. What causes microtubule dissociation from FAs? The authors propose that microtubule association leads to adhesion weakening, yet force and vinculin increase following microtubule attachment. Indeed, as vinculin increases, integrin and talin decrease early upon microtubule attachment. How can this be explained? Does increased tension expose additional vinculin binding sites, leading to a increased vinculin/talin ratio. If so, this would suggest this tension build up to be part of the regular FA maturation. Then, why are microtubules released if FAs are still stable?

3) In the last step of the proposed model, traction leads to slippage and disassembly of FAs. This is a rather vague mechanistic explanation. There are likely a number of intermediate steps leading to disassembly. Again, if the traction build-up is part of the regular FA maturation, this is known to activate FA signals via FAK triggering events leading to FA disassembly (e.g via RhoGAPs and Grb2/dynamin). This would also fit with the observation that FAK is required for microtubule associated FA turnover. The fact that CNO3 treatment does not rescue FAK inhibition suggests that FAK is downstream of Rho, which makes sense for a scenario where tension activates FAK (whereas for example PAK appears to be upstream). I don't understand what the authors mean with FAK being involved at a different stage. Again, I think these are important questions to place the observed events in the context of the known FA life cycle.

4) The quantitative modelling of the results is useful, but does not provide more insights into points raised above. Which parameters are used to model microtubule dissociation and slippage? I assume slippage is in the plots defined by reduced vinculin levels, but this is not well explained and could equally be defined by reduced integrin or talin levels. All the explanation of the modelling is rather lengthy and in my opinion could partially go to the discussion.

Minor points:

1) A Western blot comparing endogenous versus overexpressed KANK should be shown.

2) Page 5, bottom paragraph: "The focal adhesions were identified by labelled KN domain (mEmerald-KANK1-LOV2ssrA) ..." Earlier it is mentioned that the KN domain is labelled with mApple, is this a different construct?

3) Page 7, 3rd paragraph: "lose dose" should this be "low dose"?

4) Page 10, bottom: "figure 6F", I think should read "figure 6G"

5) In the Figures and legends it would make it easier if all time values would be stated either in sec or min, but not mixed.

6) Fig.2D, bottom right panel: From this image it is to me not obvious that SiR-Tubulin levels are down at time 4:10.

7) Fig.3F: " yellow dotted line" I think should read "blue dotted line"

8) Fig.5C and Suppl.Fig.3: It would be nice to plot in the supplementary figure the positive and negative hits together in one graph

Referee #1:

The manuscript by Aureille and colleagues is an extension of a previously published paper from the Bershadsky lab (reference 17). The authors state on page 12 that 'The key observation was that the initial increase of the number of microtubules associated with focal adhesion was followed by withdrawal of microtubules from the focal adhesion zone which in turn resulted in sliding and gradual disassembly of the focal adhesion.' I agree with this conclusion. However, the authors should include more control experiments to ensure that their conclusion is correct.

We are grateful to Prof Fässler for careful reviewing of our manuscript and suggestions for its improvement.

(1) HT1080 cells express high levels of KANK2 and lower levels of KANK1, both of which have not been depleted (or even better Crisp/Cas9-deleted) prior to the overexpression of the KANK1 transgene. If this experiment is not carried out it is difficult to exclude that the reported biology is simply due to overexpression of KANK1. We agree with this statement and performed the double siRNA knockdown of KANK1 and KANK2 in HT1080 cells before transfection of these cells with OptoKANK constructs. We have found that the effect of OptoKANK expression and activation on cells with depleted KANK1/2 was exactly the same as on control cells. Our western blot data show that expression of endogenous KANK1 and KANK2 after siRNA knockdown was reduced more than 90% in these experiments. We included these results in the revised manuscript (**Supplemental Figures 1G and 1H**).

(2) The authors fail to demonstrate to what extent they have overexpressed KANK1. I cannot find a quantification of the overexpressed KANK1 (and the endogenous KANK1 and KANK2).

Indeed, our data show that OptoKANK constructs were overexpressed as compared to endogenous KANK1 and KANK2. This can be inferred from the observation that cells expressing OptoKANK demonstrate increased size of focal adhesions as compared to control cells (**see Supplemental Figure 1C in the revised manuscript**). This means that OptoKANK constructs displaced the endogenous KANK1 and KANK2 molecules from their normal localizations and therefore uncoupled microtubules from focal adhesions similarly to KANK1/2 deletion mutants, as we published in the previous paper (Rafiq, N.B.M. *et al.* 2019). In fact, the focal adhesion phenotype of cells overexpressing these deletion mutants, as well as OptoKANK constructs, resembles those of cells with KANK1/2 knockdown. Of course, we agree that it is important to estimate the level of OptoKANK overexpression directly. Our antibody to KANK1 recognizes SSpB- Δ KN-mEmerald, which permitted the rough estimation of the degree of SSpB- Δ KN-mEmerald expression as compared to endogenous KANK1 protein. This comparison revealed that the level the SSpB- Δ KN-mEmerald, was

comparable with that of endogenous. We included these results in the revised manuscript (**Supplemental Figure 1A**).

(3) Furthermore, overexpression of KANKs over a certain threshold leads to cell death. It is possible that this does not happen with the optogenetically regulated fusion of two parts of KANK1 into one, and not in HT1080 cells. However, it should be excluded. We used western blot of activated (cleaved) caspase 3 as an apoptosis marker following the western blotting protocol provided by the manufacturer (**Supplemental Figure 1B**). We have found that OptoKANK expression in WT H1080 cells containing endogenous KANKs did not induce caspase 3 activation. At the same time, control experiment with incubation of these cells with 70% ethanol for 20 min (standard protocol for the induction of cell apoptosis) reveal a strong activation of caspase 3. This experiment is now included in the revised manuscript (**Supplemental Figure 1B**).

(3) KANKs can homo- and heterodimerize. Hence, the transgene can form heterodimers with the endogenous KANK1 and KANK2. Whether this happens and how this possibility affects the transgene before and after illumination is not addressed by the authors.

We believe that experiment with expression of OptoKANK in cells with depleted KANK1/2 mentioned in the answer to question (1) shows that the effect of OptoKANK does not depend on heterodimerization of OptoKANK constructs with endogenous KANK1 or KANK2.

(4) Finally, the authors claim that KANK links Talin with the CMSC. However, this has never been shown. It would be fantastic if they could show that a single photogenetically fused KANK1 molecular binds talin on the N-terminal side and Liprin/Kif on the C-terminal side. This would strengthen their claim that KANK links FAs and MTs and would exclude that KANK is simply present in two different locations that are accidentally in proximity to each other.

Indeed, the existence of the ternary complexes between KANK, talin and liprin β (or Kif21) was not yet demonstrated. However, our data published previously clearly showed that preferential targeting of microtubules to focal adhesions measured by localization of microtubule tips, depends on KANK1 and KANK2 (Rafiq *et al.* 2019 – figure 2). Moreover, in our present paper we have demonstrated that activation of OptoKANK by blue light illumination increased the number of microtubule tips overlapping with focal adhesions after illumination starts. Altogether, these data show that focal adhesion localization of endogenous KANK1 or of KN and Δ KN parts of OptoKANK connected with each other by illumination increased the probability of microtubule tips localization in the proximity of these adhesions. The exact molecular mechanism of this effect requires further studies which are beyond the scope of the present work. We made now the changes in the introduction and emphasized that, in

agreement with Prof Reinhard Fässler's comment, the statements that KANK physically links Liprin β (or Kif21) with talin is not directly proven yet.

It is also interesting to note that the authors discuss our paper (reference 18) and conclude that the KANK2 mechanism described in reference 18 does not help explaining the findings of their manuscript. Apparently, it even contradicts their findings. We used fibroblasts that were depleted of endogenous KANK2, plated the fibroblasts on FN only and analyzed FA belt slide, which precedes FA disassembly. We did not analyze in reference 18 the role of KANK in FA disassembly. I also have difficulties to understand why a failure to 'reduce podosome numbers upon KANK1 overexpression (is) inconsistent with the idea that KANK2 mediates weakening of talin (ABS2)-actin interaction'? Podosomes express KANK1 and low levels of KANK2, at least in vivo (<https://doi.org/10.1016/j.yexcr.2020.112391>). The claim of the KANK1 podosome connection is again based on overexpression and has little to do with FA belt sliding. I think Bershadsky and colleagues should read reference 18 more carefully and take the different experimental design and aims between the papers into account when they compare results and outcome.

We are very grateful to Prof Reinhard Fässler for clarification of his point of view concerning the relationship between his data and the phenomenon described in our paper. We do not see any contradiction between the results of these two studies. Our work shows that OptoKANK-induced sliding and disassembly of focal adhesions is a microtubule dependent phenomenon. Activation of OptoKANK in cells treated with nocodazole and lacking microtubules did not induce focal adhesion sliding and disassembly in our experiments. At the same time, we agree with Prof Fässler that our discussion of the possible mechanism of focal adhesion sliding upon activation of OptoKANK was premature and decided to withdraw it from the revised version of the paper. We thank Prof Fässler for his comments.

Referee #2:

This manuscript describes the generation and application of an optogenetic probe that enables modulating the activation of the microtubule binding protein KANK1 with high spatio-temporal control in cells. KANK1 has been previously shown to regulate integrin-mediated cell adhesion through its interaction with talin, an effect that is sensitive to the presence of the RhoA activator GEF-H1. However, the molecular details underlying this phenomenon are not fully understood. In this study, the author established a system to study KANK function using optogenetics. For this, the KANK protein is split in two parts: The first part is fused to LOV2SsrA, while the second part is attached to SSpB. Since LOV2ssrA and SSpB undergo dimerization upon light illumination at 488 nm, the KANK-mediated activities at focal adhesions (FAs) can be

tightly controlled and studied using live cell imaging. The results demonstrate that KANK1-mediated recruitment of microtubules leads to a local increase in myosin-II activity, which requires GEF-H1 and the presence of FAK, PAK, Kinesin-1 and α -TAT. Computer simulations support a model in which microtubule detachment from FAs induces GEF-H1 release triggering myosin activation and FA sliding ultimately leading to FA disassembly. Overall, the study is very carefully conducted, the data are of high quality and the manuscript is well written. The described effects seem to be consistent with previously proposed mechanisms and the identification of new modulators advance our understanding of microtubule/KANK-mediated control of FA turnover. I have a few minor comments but, in general, recommend the publication of this study.

We are grateful to this Reviewer for overall positive evaluation of our paper.

1. The here presented experiments use an overexpression system, where KANK constructs are expressed on top of the endogenously expressed KANK proteins. I feel that the study would benefit from characterizing the overexpression effects in more detail. For instance, the authors acknowledge in the discussion that FAs in cells expressing the OptoKANK constructs are "significantly larger" than control cells. It would be helpful if this increase in FA size as well as the degree of KANK overexpression were quantified. Which KANK isoforms are endogenously expressed in HT1080 cells? It would be also worth clarifying whether the overall morphology and structure of the actomyosin network are altered compared to the control situation, and whether the degree of FAK activation (pY397 or pY567/577) is affected by the OptoKANK expression. While these experiments will not change the overall outcome of the study, it will help the reader to interpret the experiments.

These questions are essentially similar to the questions by Reinhard Fässler which we answered above. Briefly, we succeeded to reproduce the main phenomenon, sliding and disassembly of focal adhesions upon optogenetic targeting of microtubules, in cells with almost complete knockdown of endogenous KANK1/2. Our estimation of the level of expression of the exogenous SSpB- \$\Delta\$ KN-mEmerald construct using the antibody to KANK1, shows that it was comparable to the endogenous KANK1 level. The expression of mApple-KN-LOV2ssrA was estimated by the staining with antibody to mApple and therefore cannot be compared with the expression of endogenous KANK1 visualized by a different antibody. Nevertheless, the expression of KN should not strongly exceed the expression of \$\Delta\$ KN, because otherwise the optogenetic activation of the coupling between KN and \$\Delta\$ KN would not restore the link between microtubule and focal adhesion due to competition between the coupled KN- \$\Delta\$ KN construct and the excess of KN. The level of expression of the exogenous OptoKANK constructs was sufficient to compete with endogenous KANK1 function and promote disconnection of microtubules from focal adhesions resulting in focal adhesion enlargement in agreement with our previous results (Rafiq *et al.* 2019). Indeed, our measurements that we now included in the revised version of the manuscript show

that average size of focal adhesions in cells expressing OptoKANK is larger than in control cells (**Supplemental Figure 1C**). Concerning the phosphorylation of FAK, we checked the level of pY397 using corresponding antibodies and found that it was somewhat higher than control, in both KANK1/2 knockdown cells and in these cells expressing OptoKANK construct (**Supplemental Figure 1D**). Finally, expression of OptoKANK construct in our experiments did not induce apoptosis in HT1080 cells as we demonstrated using caspase 3 activation assay (**Supplemental Figure 1B**).

2. The experiments seem to focus on one type of adhesion structure, namely comparably large, peripheral FAs at the cell edge. An exciting aspect of here presented technique is that distinct FAs classes can be studied. Fig. 2A or Fig. 3A, for instance, show that KANK-KN is also found in centrally located adhesion structures. Does an illumination at these sites also lead to FA sliding and disassembly, as observed for peripheral FAs? Expanding the experiment in this way could increase the impact of the study.

This is an interesting question. We did not address it previously because while some HT1080 cells indeed demonstrate singular adhesions in the central area, vast majority of these cells have only peripheral focal adhesions. We have now examined the effect of OptoKANK activation on centrally located focal adhesions and found that they are less sensitive to such treatment. In some cases, the illumination promoted the slow sliding of such adhesions but not their disassembly during the period of observation. Even though in depths investigation of the difference between peripheral and central adhesions is beyond the scope of the present study, we have now included these results in the supplementary data (**supp. figure 1E and F.**).

3. All images lack scale bars, please include them. Western blots should indicate molecular weights in main figures and supplementary figure 6. To allow replication of the data on other microscope setups, please indicate the laser intensity that was used for optogenetic modulation in mW/area.

We are thankful for these comments and have now corrected the manuscript accordingly.

4. It is often unclear, which statistical test was used to evaluate experimental results. In contrast to the statement in "Statistical analyses", I could not find this information in the figure legends of Fig. 3E, Fig. 4A, C, Fig. 5A-C.

The revised manuscript is now corrected accordingly.

The technological approach is conceptually highly similar to a previous study, in which light modulation of the LOV2SsrA-SspB dimer is used to study the mechanics of talin-

1 (Yu et al. Phys Rev X, 2020). I could not find the reference to this study and recommend to include it.

We are grateful for this comment and included the reference to this paper in the text.

Referee #3:

Summary:

In this study the authors investigate how microtubule attachment to focal adhesions (FAs) via KANK affects focal adhesion dynamics. Employing a photoinducible system they are able to attract microtubule tips to focal adhesions and then use this system to study how this triggers a sequence of events eventually leading to focal adhesion disassembly. Based on their study they propose a model where microtubule dissociation from focal adhesions leads to release of GEF-H1, activation of the Rho-ROCK-myosin II cascade, increased tension, sliding and finally focal adhesion disassembly. The results are solid and provide important new insights into the role of microtubules in focal adhesion dynamics and the mechanisms involved. The data and novel mechanistic insights are very valuable, however, some of the interpretations need to be reevaluated and placed into the context of the regular FA life cycle.

We are grateful to this Reviewer for kind appreciation of importance of our results.

Main points:

1) The authors present microtubule attachment as a mechanism of focal adhesion disassembly and microtubules have generally been ascribed to this role. However, the results presented in this work demonstrate the occurrence of a number of events prior to FA disassembly. Since microtubule attraction is studied in this work only for already established FAs, it is difficult to place these events along the progression of FA maturation and disassembly. It is therefore unclear whether the increase in tension following microtubule disassembly is indeed a specific signal for FA disassembly or could it be part of the regular FA maturation and traction build-up required for motility? To answer this question, it would be important to know at what stage of the FA life cycle microtubules attach to FAs. Does this only occur in late stage FAs?

The Reviewer raised an interesting and important question which perhaps require substantial further work. In this study, we did not investigate in details the microtubules collision with focal adhesions in the course of focal adhesion life cycle. Usually, we selected for our experiments well developed mature focal adhesions and therefore cannot say how forced targeting of microtubules will affect smaller/rapidly growing focal adhesions. We feel that such detailed investigation is beyond the scope of the

present study. In many cases, when focal adhesions are formed at the cell leading edge, they undergo maturation earlier than microtubule ends approach them. Thus, our optogenetic targeting of microtubules to mature adhesions most probably reflects the physiological regulation. At the same time, we agree that response of the focal adhesions to microtubule targeting could depend on their status (the stage of their life cycle and/or microenvironment). Indeed, we have now noticed (addressing question of Reviewer #2) that centrally located focal adhesions are less sensitive to microtubule targeting than the peripheral ones. We have now included this result in our revised manuscript (**supp figures 1E and 1F**). Obviously, the details of the microtubule-dependent feedback regulation of the focal adhesion turnover deserve further studies.

2) One step in the proposed sequential model is not mechanistically explained. What causes microtubule dissociation from FAs? The authors propose that microtubule association leads to adhesion weakening, yet force and vinculin increase following microtubule attachment. Indeed, as vinculin increases, integrin and talin decrease early upon microtubule attachment. How can this be explained? Does increased tension expose additional vinculin binding sites, leading to an increased vinculin/talin ratio. If so, this would suggest this tension build up to be part of the regular FA maturation. Then, why are microtubules released if FAs are still stable?

First, let us note that vinculin increase is very mild and transient, and in any case, much less significant than the pronounced and steady decrease of vinculin signal (after the delay) after illumination. Immediate steady integrin beta3 and talin decrease after the illumination is a clear indication that one of the early responses is the adhesion weakening, while vinculin transiently persists. Next, microtubule release, on the microscopic molecular scale, probably cannot sense the level of the FA stability, and is rather dependent on specific molecular contacts. Some studies indicate that a contact of microtubules with the focal adhesions promotes microtubule catastrophes via a specific biochemical mechanism (Efimov, A. and Kaverina, I., 2009). We posit, based on a number of published studies, that KANK is one of the main such contacts, and in our model the explanation for the accelerated microtubule dissociation from the FA upon the illumination is very straightforward: illumination rapidly increases KANK, which brings more microtubules (confirmed by our data), but these microtubules bring with them molecular agents weakening the FA (promoting talin/integrin dissociation). As talin is one of the main contact point between the FA and KANK, this rapidly leads to KANK signal reversal – KANK starts to dissociate, which in turn decreases the pause of microtubules on the FA before the microtubules switch to shortening, and this effectively leads to the microtubules steadily leaving the FA. Our quantitative modeling confirms the plausibility of this scenario – these very likely feedbacks collectively lead to the observed and modeled brief transient increase, and then steady significant decrease of the number of microtubules in contact with the FA.

3) In the last step of the proposed model, traction leads to slippage and disassembly of FAs. This is a rather vague mechanistic explanation. There are likely a number of intermediate steps leading to disassembly. Again, if the traction build-up is part of the regular FA maturation, this is known to activate FA signals via FAK triggering events leading to FA disassembly (e.g via RhoGAPs and Grb2/dynamin). This would also fit with the observation that FAK is required for microtubule associated FA turnover. The fact that CNO3 treatment does not rescue FAK inhibition suggests that FAK is downstream of Rho, which makes sense for a scenario where tension activates FAK (whereas for example PAK appears to be upstream). I don't understand what the authors mean with FAK being involved at a different stage. Again, I think these are important questions to place the observed events in the context of the known FA life cycle.

First, we emphasize that it's not only traction that leads to the slippage, but the traction combined with weakened adhesiveness – we would argue that this is specific, not vague, mechanistic explanation. Second, we only model the slippage of the FA, not the subsequent FA disassembly, which indeed is a much more complex process. Specifically, we agree that the force balance in the FA which is already sliding could be more delicate. We assume that after the onset of sliding induced by OptoKANK activation, the FA could experience only weak pulling force. Under the condition of low pulling force, the FA undergoes disassembly similarly to the situation of total inhibition of myosin activity. Of course, sliding *per se* is not always correlated with focal adhesion disassembly. For example, we noticed that the OptoKANK activation at centrally-located focal adhesions induced their slow sliding but not disassembly. It might be that for such adhesions the traction force decrease during sliding is smaller than for peripheral focal adhesions and therefore insufficient to trigger the disassembly. We understand that this issue requires further studies and mentioned it in the discussion.

Concerning the functions of FAK, we agree that data from literature suggest that it can participate in focal adhesion disassembly. We have now introduced the corresponding reference into the revised version of the manuscript. However, we do not think that FAK works in concert with dynamin because inhibition of dynamin by dynasore and hydroxydynasore, which inhibit endocytosis was not sufficient to prevent the effect of OptoKANK activation. We have now included these considerations into the Discussion.

4) The quantitative modelling of the results is useful, but does not provide more insights into points raised above. Which parameters are used to model microtubule dissociation and slippage? I assume slippage is in the plots defined by reduced vinculin levels, but this is not well explained and could equally be defined by reduced

integrin or talin levels. All the explanation of the modelling is rather lengthy and, in my opinion, could partially go to the discussion.

We appreciate the Reviewer's opinion about the usefulness of modeling. The modeling offers two main insights: 1) even when condensed and lumped together, the number of essential pathways involved in the microtubule-directed process of FA weakening makes it too hard to just intuit whether these pathways could explain the order of molecular and mechanical events observed. The simulations can verify the plausibility of the model. 2) Only quantitative modeling can explain how GEF-H1 can increase near FA.

As for the insights about the microtubule release and slippage; To answer the reviewer's question about modeling microtubule dissociation: it is a function of engaged KANK level on the FA (see equation 1 of the supplemental material and explanations in the 2 paragraphs that follow that equation).

Slippage in the plots is not defined by the reduced vinculin level. Quite the opposite, the reduced vinculin level, according to the model, is largely explained by the slippage and shrinking of the adhesion. The slippage in the model is explained as the combined effect of the increasing actomyosin pulling force and weakening adhesion (stick-slip transition). Effective slippage velocity is known to be proportional to the ratio of the force to the effective adhesiveness. When this ratio exceeds a threshold, the stick-slip transition occurs, and the slippage starts. And the Reviewer is exactly right – the slippage is defined by the reduced integrin and talin levels – this is even in the mathematical equations! To answer the Reviewer's question: mathematically, the slippage is defined by the ratio of the active pulling myosin to the talin level (see equations 7-9 of the supplemental material).

We appreciate the advice about the lengthy explanations. We followed this advice, and moved a part of the modeling section into the Discussion.

Minor points:

1) A Western blot comparing endogenous versus overexpressed KANK should be shown.

We have now showed these data in the **Supplemental Figure 1A**.

2) Page 5, bottom paragraph: "The focal adhesions were identified by labelled KN domain (mEmerald-KANK1-LOV2ssrA) ..." Earlier it is mentioned that the KN domain is labelled with mApple, is this a different construct?

Corrected.

3) Page 7, 3rd paragraph: "lose dose" should this be "low dose"?

Corrected.

4) Page 10, bottom: "figure 6F", I think should read "figure 6G"

Corrected.

5) In the Figures and legends it would make it easier if all time values would be stated either in sec or min, but not mixed.

Corrected.

6) Fig.2D, bottom right panel: From this image it is to me not obvious that SiR-Tubulin levels are down at time 4:10.

Indeed, in the figure 2D, the focal adhesion circled at 4 min 10 sec is still partially overlapped with microtubules. The purpose of this figure was to demonstrate the overlapping of microtubules and focal adhesions induced by OptoKANK activation. Eventually, however the focal adhesion area is becoming completely free of microtubules as shown in a **supplementary movie 5** which we have now included into the revised manuscript. The quantification shown in the graph 2E reflects the statistical data based on measurements on 18 adhesions from 2 independent experiments.

7) Fig.3F: " yellow dotted line" I think should read "blue dotted line"

Corrected.

8) Fig.5C and Suppl.Fig.3: It would be nice to plot in the supplementary figure the positive and negative hits together in one graph

We have updated the revised version of the manuscript accordingly.

Dear Julien,

Thank you for submitting a revised version of your manuscript. Your study has now been seen by all original referees, who find that most of their previous concerns have been addressed and now recommend acceptance of the manuscript. There now remain only a few, mainly editorial, editorial points that need addressing before I can extend formal acceptance of the manuscript:

1. Please address the remaining minor comments from the reviewers.
2. Please list authors' names following the convention "first name, initial, last name" in the manuscript text, similarly as on our website.
3. Please add your institutional email address in our system.
4. Please submit a complete author checklist, which you can download from our author guidelines (<https://www.embopress.org/pb-assets/embo-site/EMBO%20Press%20Author%20Checklist-1642513524327.xlsx>). Please insert information in the checklist that is also reflected in the manuscript. The completed author checklist will also be part of the Review Process File.
5. Please submit up to five keywords.
6. Please upload the main figures as individual production quality figure files in the .eps, .tif, or .jpg format (one file per figure) and remove figures from the manuscript file.
7. Please make sure that the order of the sections in the manuscript is as follows: abstract, introduction, results, discussion, materials & methods, data availability section, acknowledgments, disclosure statement and competing interests, references, main figure legends, tables, expanded figure legends.
8. Please compile the Appendix figures together with their legends and the rest of the supplementary information into a single PDF file labelled "Appendix". Please preface the Appendix with a brief table of contents. The pages should be numbered. Please update the nomenclature of Appendix figures to Appendix Figure S1 etc.
9. Alternatively, we can accommodate up to five Expanded View (EV) figures that are collapsible/expandable online. EV Figures should be cited as "Figure EV1, Figure EV2" etc. in the text and their respective legends should be included in the main text after the legends of regular figures. They should also be uploaded as individual, production quality figure files, similar to the main figures. Further information on the format is available here:
<https://www.embopress.org/page/journal/14602075/authorguide#expandedview>.
10. Please split the Western blot source data currently found in the Appendix figure S7 into a single file/folder per figure, to be uploaded as "source data" for these figures.
11. CRediT has replaced the traditional author contributions section because it offers a systematic, machine-readable author contributions format that allows for more effective research assessment. Please remove the Authors Contributions from the manuscript and use the free text boxes beneath each contributing author's name in our online submission system to add specific details on the author's contribution. More information is available in our guide to authors.
12. Please rename "Conflict of interest" section into "Disclosure and competing interests statement" (further info: <https://www.embopress.org/page/journal/14602075/authorguide#conflictsofinterest>).
13. Please update references according to The EMBO Journal style - where there are more than 10 authors on a paper, the first 10 should be listed, followed by 'et al.' Please see further information here:
<https://www.embopress.org/page/journal/14602075/authorguide#referencesformat>
14. Please rename the movies into Movie EV1-EV10 and update the callouts accordingly. The legends should be removed from the manuscript text file and zipped with each movie file. Further information is available here:
<https://www.embopress.org/page/journal/14602075/authorguide#expandedview>
15. In the Data Availability section, please add a resolvable URL to the BioStudies dataset.
16. The following figures are not mentioned in the manuscript text: Figure 6E, Appendix Figure S5, Appendix Figure S7.
17. During our standard image integrity check, we noticed re-use of Western blot panels in Figure 4B and Figure 5A. Please check and either correct or clearly state this in the figure legend.
18. Our data editors have flagged the following issues in figure legends that need correcting:
 - Please define the annotated p values */*** in the legend of figure 2a, supplementary figure 4a as appropriate.
 - Please indicate the statistical test used for data analysis in the legends of figure 2a, supplementary figure 4a.
 - Please note that the box plots need to be defined in terms of minima, maxima, centre, bounds of box and whiskers, and percentile in the legends of figures 4c; 5a-c, supplementary figure 4a.
 - Please note that information about the number and nature of the replicates is missing in the legends of figure 2c, supplementary figure 2b.
 - Please describe the nature of replicates in the legends of figures 1c-d; 2e; 4c; 5a-c, supplementary figure 1f-g; 3a; 4a.
 - Please define the error bars in the legends of figure 2c, supplementary figure 2b.
 - Please define the scale bar for figures 2a-b, d; 3a, e-f; 4a-c; 5a-c, supplementary figure 1c, e, h; 2a-c; 3a; 4a; 5a.
 - Please define the white dotted line in the legends of figures 2d; 3e.
 - Please define the blue dotted line in the legend of supplementary figure 1e.
 - Please define the white and yellow dotted lines in the legend of supplementary figure 2b.
 - Please note that the blue dotted line in figure 4c is mentioned as yellow dotted line in the legend. This needs to be rectified.
19. Papers published in The EMBO Journal are accompanied online by a 'Synopsis' to enhance discoverability of the

manuscript. It consists of A) a short (1-2 sentences) summary of the findings and their significance, B) 3-4 bullet points highlighting key results and C) a synopsis image that is 550x300-600 pixels large (width x height, jpeg or png format). You can either show a model or key data in the synopsis image. Please note that the image size is rather small and that text needs to be readable at the final size. Please send us this information together with the revised manuscript.

With best wishes,

leva

leva Gailite, PhD
Senior Scientific Editor
The EMBO Journal
Meyerhofstrasse 1
D-69117 Heidelberg
Tel: +4962218891309
i.gailite@embojournal.org

We realize that it is difficult to revise to a specific deadline. In the interest of protecting the conceptual advance provided by the work, we recommend a revision within 3 months (6th Jun 2024). Please discuss the revision progress ahead of this time with the editor if you require more time to complete the revisions.

Referee #1:

the authors carefully revised the manuscript. Their revisions are very fine and it is ready for publication in EMBO J. I only think that one sentence in the introduction is not logic:

'While the existence of ternary complexes between talin, KANK and liprin- β 1 or between talin, KANK and KIF21A was not directly demonstrated, it was shown that knockdown of KANK1 and KANK2 abolished the microtubule targeting to focal adhesions¹⁷. Thus, it is reasonable to suggest that KANK can directly or through the liprins-ELKS-LL5 β complex couple microtubule ends with the focal adhesions^{1, 17-19}.'

How should a MT defect in the absence of KANK1 or KANK2 justify the conclusion that KANK1 links CMSC and FA and exclude that the two KANKs work as two independent pools in these two complexes? It is possible that the KANKs link the two complexes, and in fact this would allow to turn over ECM in the vicinity of FAs. But this has just not been conclusively shown so far. Maybe it will be shown in the next paper from the Bershadsky lab.

Reinhard Fässler

Referee #2:

The authors have adjusted the main text, included additional experiments and extended the discussion. As a result, my previously raised points are addressed and I recommend the publication of this study, which enhances our current understanding of how KANK modulates focal adhesion disassembly and also provides an elegant tool to study cell adhesion turnover using optogenetics. I only have a few non-essential suggestions that the authors could follow when preparing their final manuscript.

Supp Fig. 1A, D. The densitometric analysis does not show error bars. Please include them or provide the densitometric values below the respective western blot lanes.

Supp Fig.1 H: The quantification of normalized mean intensities indicates that data in this experiment are very noisy. Can the authors run a statistical test to show that Ctrl and KANK1/2 KD are indeed different?

Fig. 6G: The figure legend for Fig. 6H should indicate what is measured on the y-axis and what is indicated by the dashed line. It also seems that the second part of the red curve (GEF-H1) is lying over another (dashed red line). Please double-check.

Referee #3:

I am overall satisfied with the explanations given. I have a few further comments.

1) The statement that focal adhesions mature prior to microtubule attachment (i.e. supporting it as a mechanism to induce disassembly) should be mentioned together with a reference/data supporting this.

2) The authors state, the fact that CNO3 does not interfere with microtubule-driven focal adhesion disassembly suggests that the involvement of FAK is not related to (or beyond) the Rho pathway. I do not agree with this. In their model, Rho activation and resulting contractility is counteracted by sliding, leading to low stretching forces and disassembly. This reversal in the effect of contractility might well be supported by other mechanisms, such as a negative feedback via FAK-p190RhoGAP. As mentioned in my previous comments, it merely suggests that FAK acts downstream of Rho. This should be clarified.

The authors addressed the remaining editorial issues.

Dear Julien,

Thank you for addressing the final editorial issues. I am now pleased to inform you that your manuscript has been accepted for publication. Congratulations on a nice paper!

Before we forward your manuscript to our publishers, I would like to propose a couple of minor changes in the article abstract and synopsis, mainly aimed at increasing the accessibility of the study to our more general audience. I have also written a short blurb that will accompany the title of your manuscript in our online table of contents. Please take a look at the text below and in the attached manuscript text file and let me know if any corrections are necessary.

Blurb:

Optogenetic targeting of KANK1 connector protein reveals the role of Rho activation-dependent actomyosin traction force in induction of focal adhesion disassembly.

Synopsis:

Progressive microtubule capture during focal adhesion maturation ultimately triggers focal adhesion turnover. This study shows that local development of actomyosin traction force induces focal adhesion sliding, thus mediating focal adhesion disassembly upon interaction with microtubules.

- An optogenetic construct of KANK1 protein enables targeting of microtubules to focal adhesions upon local blue light illumination.
- Illumination-induced increase in the number of microtubule tips at focal adhesion is followed by their withdrawal, and by accumulation of myosin-II filaments.
- Local transient development of traction force is followed by focal adhesion sliding and disassembly.
- Disassembly of focal adhesion induced by OptoKANK-driven microtubule targeting depends on Rho activation by the microtubule-associated GEF-H1 and other factors.
- A mathematical model replicates the observed microtubule-driven focal adhesion disassembly due to myosin-II mediated traction force generation.

Finally, the provided synopsis image, when resized to the required 550 pixel x 350-600 pixel dimensions becomes rather difficult to read (please find attached). Perhaps the font size could be increased to improve readability?

If you have any questions, please do not hesitate to contact the Editorial Office. Thank you again for this contribution to The EMBO Journal and congratulations on a successful publication!

Best wishes,

Ieva

Ieva Gailite, PhD
Senior Scientific Editor
The EMBO Journal
Meyerohofstrasse 1
D-69117 Heidelberg
Tel: +4962218891309
i.gailite@embojournal.org
